# Molecular composition of organic matter controls methylmercury formation in boreal lakes

Andrea G. Bravo[1], Sylvain Bouchet[2], Julie Tolu[3], Erik Björn[2], Alejandro Mateos-Rivera[1] & Stefan Bertilsson[1]

A detailed understanding of the formation of the potent neurotoxic methylmercury is needed to explain the large observed variability in methylmercury levels in aquatic systems. While it is known that organic matter interacts strongly with mercury, the role of organic matter composition in the formation of methylmercury in aquatic systems remains poorly understood. Here we show that phytoplankton-derived organic compounds enhance mercury methylation rates in boreal lake sediments through an overall increase of bacterial activity. Accordingly, *in situ* mercury methylation defines methylmercury levels in lake sediments strongly influenced by planktonic blooms. In contrast, sediments dominated by terrigenous organic matter inputs have far lower methylation rates but higher concentrations of methylmercury, suggesting that methylmercury was formed in the catchment and imported into lakes. Our findings demonstrate that the origin and molecular composition of organic matter are critical parameters to understand and predict methylmercury formation and accumulation in boreal lake sediments.

[1] Department of Ecology and Genetics, Limnology and Science for Life Laboratory, Uppsala University, SE-75236 Uppsala, Sweden. [2] Department of Chemistry, Umeå University, SE-90187 Umeå, Sweden. [3] Department of Ecology and Environmental Science, Umeå University, SE-90187 Umeå, Sweden. Correspondence and requests for materials should be addressed to A.G.B. (email: andrea.garcia@ebc.uu.se).

Humans are mostly exposed to the highly toxic methylmercury (MeHg) through fish consumption[1]. In the European Union alone, reducing environmental exposure to MeHg could save €8–9 billion per year by protecting children's brain development[2]. Unravelling the factors controlling the methylation of inorganic Hg (Hg) to MeHg in aquatic ecosystems is thus crucial for reducing such risks. Hg methylation is predominantly a microbial process mediated by some members of the sulphate-reducing bacteria, iron-reducing bacteria, methanogens and *Firmicutes*[3,4]. In aquatic systems, the process occurs in oxygen-deficient zones of sediments or stratified water columns[5,6]. A wide range of environmental factors, including temperature, pH, redox potential and the concentration of bioavailable Hg are known to influence Hg methylation processes[7]. In particular, organic matter (OM) seems to play a crucial role for MeHg formation, acting as electron donor for Hg methylating bacteria[4] while also binding Hg to regulate its bioavailability depending on the type of complexes formed[8]. In aquatic ecosystems, OM is an extremely heterogeneous mixture derived from a combination of terrestrial (allochthonous) or internal (autochthonous) sources with different microbial and physicochemical reactivity[9–11]. Because the concentration of OM alone cannot explain the variation in Hg methylation rates measured in the environment[12], it has been suggested that OM molecular composition plays an essential role in the process[13,14]. For example, low ratios of elemental carbon and nitrogen contents (C/N), being a proxy for autochthonous OM, have been associated with higher Hg methylation rates in estuarine and marine sediments[13,14]. In lakes, algae and cyanobacterial blooms have been linked to enhanced Hg methylation in sediments[15]. Moreover, it has been suggested that algal-derived compounds might be an important factor regulating the production of MeHg in periphyton[16]. Combined, these studies imply that autochthonous OM has likely a central role in Hg methylation processes in aquatic systems, but the coarse approaches used so far to characterize OM, such as loss-on-ignition[12], C/N ratios[13,14] or chlorophyll *a* measurements[16], do not distinguish between specific OM compounds that may influence Hg methylation processes in contrasting ways. To bridge this gap in knowledge, the aim of this study was to find linkages between specific organic compounds and Hg methylation in boreal lake sediments by resolving and characterizing, at the molecular level, the typically complex natural OM.

Boreal biomes contain the highest density of freshwaters on Earth[17] and host about 28.4% of the global lake area (obtained by dividing the total estimated boreal lake area[18] by the global lake area[17]). Boreal lakes are of special concern because they are highly sensitive to environmental changes such as terrigenous OM inputs and autochthonous OM production[19]. We hypothesized that, in boreal lake sediments strongly affected by different OM inputs, ranging from internally produced phytoplankton to terrigenous OM imported from the surrounding catchment, the molecular composition of the OM could be a major driver of Hg methylation. To test this, we studied a gradient of boreal lakes with contrasting OM characteristics and demonstrate that fresh phytoplankton-derived OM compounds have a strong positive influence on *in situ* Hg methylation rates. We are the first to demonstrate that Hg methylation rates can be predicted by sediment OM molecular composition.

## Results

**OM molecular composition in lake sediments.** We collected sediments from 10 boreal lakes in central Sweden (Table 1) with

**Table 1 | Characteristics of the investigated lakes.**

| Lake | Code | N | E | Area (km²) | z (m) | pH |
|---|---|---|---|---|---|---|
| Lilla Sångaren | LS | 59.8996 | 15.3923 | 0.24 | 17 | 6.9 |
| Ljustjärn | LJU | 59.92375 | 15.453472 | 0.12 | 10 | 7.3 |
| Svarttjärn | S | 59.89073 | 15.2577 | 0.07 | 6.5 | 5.6 |
| Fälaren | F | 60.33656 | 17.79396 | 2.05 | 2.0 | 7.5 |
| Oppsveten | O | 59.98874 | 15.57562 | 0.65 | 10 | 6.3 |
| Stransdjön | STR | 59.87099 | 17.168650 | 1.3 | 2.5 | 6.9 |
| Valloxen | V | 59.73846 | 17.83954 | 2.9 | 6 | 8.5 |
| Vallentunasjön | VALE | 59.50435 | 18.037083 | 5.8 | 4 | 7.1 |
| Marnästjärn | M | 60.14483 | 15.20714 | 0.15 | 2 | 7.2 |
| Lötsjön | LOTS | 59.86314 | 17.940110 | 0.63 | 7 | 6.8 |

Location, area, maximum lake depth (z) and pH. Sediment cores were collected at the maximum lake depth.

different trophic status (total phosphorus, TP: 8–198 µg l$^{-1}$) and organic carbon concentrations (dissolved organic carbon, DOC: 3.8–33.1 mg l$^{-1}$; Table 2). We characterized the molecular composition of sedimentary OM by a pyrolysis–gas chromatography–mass spectrometry (Py–GC–MS) method, recently optimized for sediments[20]. As an analytical tool to characterize OM composition, Py–GC–MS is a good compromise between the quantitative molecular information obtained from tedious, compound specific, wet chemical extractions associated with liquid chromatography (LC)–MS or GC–MS analyses, and the qualitative, non-molecular information provided by high-throughput techniques such as visible–near-infrared spectroscopy or 'RockEval' pyrolysis. In addition to the high throughput in terms of analyses and data treatment, the Py–GC–MS method used in this study yields semi-quantitative data for more than 100 pyrolytic compounds derived from organic compounds of diverse biochemical classes (for example, lignin, chlorophyll, lipids, and so on). Hence the method makes possible to explore the overall OM molecular composition and to infer the origin and degradation status of the sediment OM[20].

Differences in molecular composition of the sediment OM among the 10 lakes (Table 1) were explored using principal component analysis (PCA), performed with 110 identified pyrolytic organic compounds (Fig. 1a). The first principal component (PC1; 29% of total variance) separated autochthonous phytoplankton-derived organic compounds (Fig. 1a, positive loadings on PC1) from plant-derived organic compounds (Fig. 1a, negative loadings on PC1). Indeed lignin oligomers, specific of vascular plants[21], and phenolic compounds which derive from pyrolysis of lignin structures or originate from non-vascular plants[22] had negative loadings on PC1. Positive loadings on PC1 were found for pyrolytic products of proteins (for example, 2.5-diketopiperazines[23]) and chlorophylls (that is, phytol, phytene, phytadiene[24]), deriving both from phytoplankton-derived materials[25,26]. Pyrolytic compounds characteristic for chitin structures from, for example, fungal cell walls and arthropod exoskeletons (for example, acetamidofuran and oxazoline[27]) (Fig. 1a) also presented positive loadings on PC1. The second principal component (PC2; 26% of total variance) separated organic compounds indicative of OM degradation processes (negative loadings on PC2) from fresh plant-derived organic compounds (positive loadings on PC2). Indeed, negative loadings on PC2 were observed for pyrolytic compounds indicative of degradation products of high-molecular mass carbohydrates (for example, furans[22]); proteins and chlorophylls (for example, pyrrole, pyridine and aromatic nitriles[28]); and cell wall lipids (short-chain *n*-alkanes/alkenes with carbon number ≤13) (Fig. 1a). The mid-chain *n*-alkenes/*n*-

**Table 2 | Vertical profiles of several ancillary parameters in the studied lakes.**

| Sample | $T$ (°C) | C ($\mu$S s$^{-1}$) | $O_2$ (mg l$^{-1}$) | HgII (ng l$^{-1}$) | MeHg (ng l$^{-1}$) | MeHg (%) | DOC (mg l$^{-1}$) | TP ($\mu$g l$^{-1}$) | Chl $a$ ($\mu$g l$^{-1}$) | SUVA$_{254}$ (l mg$^{-1}$ C m$^{-1}$) | SO$_4^{2-}$ (mg l$^{-1}$) | BP ($\mu$g C l$^{-1}$ d$^{-1}$) |
|---|---|---|---|---|---|---|---|---|---|---|---|---|
| LS-WC-6 | 9.4 | 42 | 8.3 | 2.3 ± 0.1 | 0.3 ± 0.03 | 10.8 | 6.3 ± 0.2 | 10 | 4.4 | 3.3 | 3.1 | 0.18 |
| LS-WC-16 | 5.3 | 43 | 6.4 | 2.3 ± 0.0 | 0.2 ± 0.03 | 9.6 | 6.0 ± 0.5 | 10 | 0.0 | 3.8 | 3.2 | 0.15 |
| LS-OW-17 | 5.0 | 60 | 4.7 | 4.0 ± 0.1 | 0.8 ± 0.04 | 16.0 | 7.0 ± 1.0 | 23 | 1.8 | 4.2 | 2.9 | 0.10 |
| LJU-WC-2 | 16.9 | 18.8 | 9.2 | ND | ND | ND | 3.8 ± 0.1 | 8 | 2 | 1.2 | 2.1 | 1.15 |
| LJU-WC-8 | 7.8 | 16.4 | 0.9 | ND | ND | ND | 3.7 ± 0.4 | 17 | 16 | 1.3 | 2.1 | 0.84 |
| LJU-OW-10 | 6.9 | 77.2 | 0.2 | ND | ND | ND | 6.5 ± 0.3 | 96 | 93 | 1.6 | 2.2 | 0.24 |
| S-WC-1 | 15.0 | 45 | 4.7 | 7.3 ± 0.5 | 0.8 ± 0.05 | 9.5 | 26.2 ± 1.0 | 11 | 0.0 | 5 | 1.6 | 0.46 |
| S-WC-3 | 7.0 | 42 | 3.4 | 4.3 ± 0.1 | 0.5 ± 0.03 | 10.6 | 19.2 ± 2.0 | 15 | 0.0 | 5.2 | 1.9 | 0.86 |
| S-OW-6.5 | 4.8 | 59 | 0.1 | 4.6 ± 0.1 | 1.5 ± 0.05 | 25.0 | 22.0 ± 0.2 | 36 | 2.7 | 6 | 0.9 | 0.49 |
| F-WC-1 | 18.7 | 67 | 8.8 | 2.7 ± 0.1 | 0.5 ± 0.02 | 15.3 | 33.1 ± 1.0 | 23 | 31.5 | 3.9 | 3.4 | 4.27 |
| F-OW-2 | 17.6 | 67 | 8.6 | 2.8 ± 0.1 | 0.3 ± 0.02 | 10.1 | 32.6 ± 0.8 | 20 | 8.9 | 3.9 | 3.0 | 0.13 |
| O-WC-4 | 17.4 | 26 | 8.7 | 5.2 ± 2.6 | 0.5 ± 0.11 | 8.9 | 18.8 ± 0.3 | 13 | 0.9 | 4.3 | 2.1 | 0.80 |
| O-WC-9 | 8.5 | 30 | 4.7 | 3.4 ± 0.1 | 0.5 ± 0.04 | 12.1 | 17.1 ± 0.0 | 19 | 0.9 | 4.6 | 2.4 | 0.28 |
| O-OW-10 | 8.6 | 30 | 0.8 | 6.3 ± 0.1 | 0.5 ± 0.01 | 7.9 | 16.7 ± 1.4 | 14 | 0.0 | 4.8 | 2.3 | 0.26 |
| STR-WC-1 | 16.4 | 140.1 | 8.5 | ND | ND | ND | 18.8 ± 0.6 | 34 | 10.8 | 3.2 | 4.7 | 1.63 |
| STR-OW-2.5 | 16.4 | 285 | 0.3 | ND | ND | ND | 19.6 ± 0.6 | 60 | 13.1 | 3 | 4.6 | 1.75 |
| V-WC-2 | 19.7 | 338 | 9.8 | 1.2 ± 0.1 | 0.2 ± 0.06 | 16.5 | 14.9 ± 2.3 | 30 | 52.4 | 2.2 | 8.9 | 4.58 |
| V-OW-6 | 18.8 | 502 | 0.1 | 0.9 ± 0.1 | 0.3 ± 0.01 | 23.0 | 12.3 ± 0.1 | 49 | 52.4 | 2.5 | 8.9 | 3.88 |
| VALE-WC-1 | 17.2 | 331 | 8.6 | ND | ND | ND | 13.3 ± 0.4 | 77 | 62.8 | 1.5 | 16.2 | 3.18 |
| VALE-OW-4 | 17.2 | 469 | 0.2 | ND | ND | ND | 14.0 ± 0.3 | 77 | 57.8 | 1.4 | 16.2 | 3.92 |
| M-WC-1 | 17.8 | 185 | 6.3 | 2.4 ± 0.2 | 2.8 ± 0.30 | 53.8 | 9.6 ± 0.5 | 198 | 171 | 1.7 | 3.5 | 8.12 |
| M-OW-2 | 17.8 | 185 | 6.3 | 1.6 ± 0.1 | 2.9 ± 0.10 | 64.4 | 9.2 ± 0.7 | 185 | 190 | 1.8 | 3.6 | 6.57 |
| LOTS-WC-2 | 18.2 | 207.1 | 9.1 | ND | ND | ND | 11.9 ± 0.7 | 21 | 8.6 | 1.8 | 3.8 | 0.75 |
| LOTS-WC-6 | 17.4 | 209.9 | 5.5 | ND | ND | ND | 13.2 ± 0.7 | 16 | 7.5 | 1.7 | 3.8 | 0.60 |
| LOTS-OW-7 | 11.5 | 288 | 0.3 | ND | ND | ND | 13.3 ± 0.7 | 65 | 18.1 | 1.7 | 1.6 | 0.61 |

F, Fälaren; LJU, Ljustjärn; LOTS, Lötsjön; LS, Lilla Sångaren; M, Marnästjärn; ND, not determined; O, Oppsveten; S, Svarttjärn; STR, Strandsjön; V, Valloxen; VALE, Vallentunasjön.
Sample codes refer to: Lake Code-sample type (that is, WC (water column) or OW (water overlying the sediment))-depth, for example, LS-WC-6 refers to Lilla Sångaren, water column sample at 6 m depth.

alkanes C22–24 known to originate from the pyrolysis of resistant bio-macromolecules such as fungal/animal chitin (for example, acetamidofuran) and plant cutin/suberin or algaenan[28], had also negative loadings on PC2 (Fig. 1a). In contrast, pyrolytic products of high-molecular mass carbohydrates or polysaccharides (that is, levosugars[22]) and of long-chain cell wall lipids (that is, $n$-alkanes C27–C29 and alkan-2-ones C29–C33) deriving from plant material, and known to be available for bacterial communities[21], had positive loadings on PC1.

The studied lakes exhibited a large variation in OM molecular composition as demonstrated by the scatter of sediment samples within the ordination space (Fig. 1b). Sediments from Lötsjön (LOTS), Marnästjärn (M), Strandsjön (STR), Vallentunasjön (VALE) and Valloxen (V) were dominated by autochthonous OM (positive scores on PC1). Whereas LOTS, M, VALE and V were rich in fresh phytoplankton-derived OM (proteins, chlorophyll), STR contained high proportions of chitin-derived and nitrogen containing organic compounds arising with degradation processes of chlorophyll and proteins (negative scores on PC2) (Fig. 1b). This suggests high invertebrate presence in STR as reported elsewhere[29], and more extensively degraded sedimentary phytoplankton-derived OM in comparison to the other 4 lakes dominated by autochthonous OM. On the other hand, sediments from Svarttjärn (S), Ljustjärn (LJU), Lilla Sångaren (LS), Oppsveten (O) and Fälaren (F)

were dominated by allochthonous OM of terrestrial origin (negative scores on PC1). More specifically, S and LJU were enriched in recently produced plant-derived OM compounds (positive scores on PC2) while LS and O contained degraded OM compounds (negative scores on PC2). The molecular composition of OM in F sediments was not well explained by either of the two principal components (low PC-scores; Fig. 1b) but featured high proportions of lignin- and phenolic-derived compounds, indicative of plant-derived OM, and low proportions of proteins and chlorophyll (Supplementary Table 3). The molecular composition of OM in these sediments thus indicates dominance of terrigenous OM but with a composition that differs from S, LJU, LS and O. It is noteworthy that the two sediment depth layers (0–1 and 1–2 cm) from each individual lake plotted close to each other (Fig. 1b) within the PCA ordination space. Indeed we did not find a clear pattern for organic compounds known to be rapidly degraded (that is, levosugars[22]; chlorophylls[24] and proteins[30]) (Supplementary Table 4) between the two sediment depth layers, implying homogeneity of sedimentary OM molecular composition within the top two cm of the studied boreal lake sediments.

**Predicting Hg methylation with OM molecular composition.** Potential Hg methylation and MeHg demethylation rate constants were determined in sediments using enriched isotope tracers

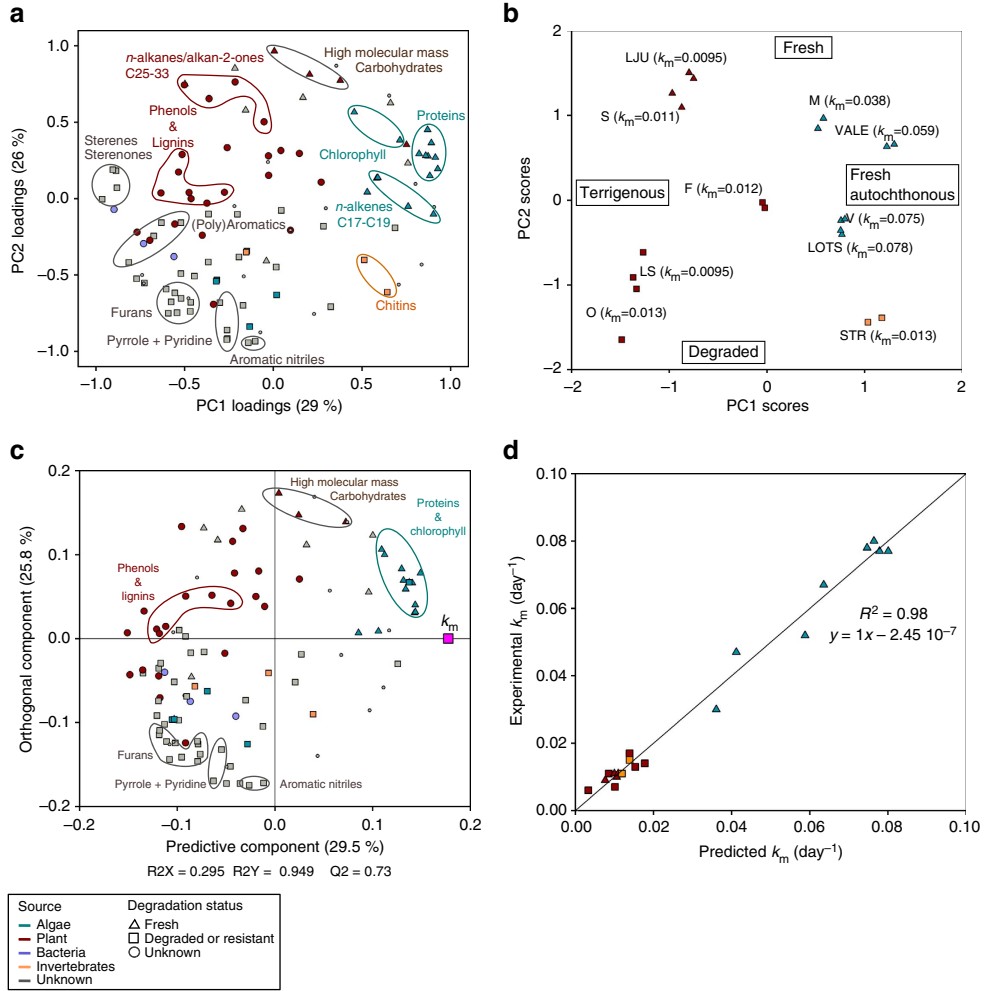

**Figure 1 | OM molecular composition and Hg methylation rate constants.** PCA (**a,b**) and orthogonal projections to latent structures statistical model (OPLS) for $k_m$ (**c,d**). The pyrolytic organic compounds were sorted out into five categories according to their origin: algae, plant, bacteria, invertebrate or unknown); and three categories according to degradation status: fresh, degraded or unknown (for example, gray circles correspond to compounds with unknown origin (that is, gray) and unknown degradation status (that is, circle)). (**a**) PC1/2-loadings (**b**) PC1/2-scores and Hg methylation rate constants ($k_m$) (**c**) loadings of OM compounds with a predictive (predictive component) and not predictive capacity (orthogonal component) of the OPLS model for $k_m$ (**d**) experimental $k_m$ values versus $k_m$ values predicted by the OPLS modelling.

(Supplementary Table 2). Hg methylation rate constants ($k_m$) were highest in lakes where the OM was dominated by fresh phytoplankton-derived compounds (LOTS, M, VALE and V; 0.038–0.075 day$^{-1}$, Fig. 1b). Methylation rate constants were dramatically lower in sediments that were either enriched in terrigenous compounds (S, LJU, LS, O and F; 0.0095–0.013 day$^{-1}$) or in sediments with high contributions of invertebrate chitin and degraded phytoplankton-derived residues (STR; 0.013 day$^{-1}$). An orthogonal projections to latent structures (OPLS, model I) statistical model was developed to explain and predict $k_m$ from the detailed molecular composition of sediment OM (Fig. 1c). OPLS is a recent modification to the PLS regression analysis method that separates the systematic variation of the OM molecular composition ($X$) into two types of components, that is, predictive components which are linearly related to $Y$ (here $k_m$) and orthogonal components to $Y$ ($k_m$)[31]. Accordingly, the predictive component of the OPLS model I represents the compounds directly correlating with $k_m$ whereas the orthogonal component denotes the compounds that are not related to $k_m$. The $R^2X$ value indicates the proportion of variance in the $X$ variables (that is, the pyrolytic organic compounds) explained by the model, while the $R^2Y$ value indicates the proportion of

variance in $k_m$ explained by the model. With one predictive component ($R^2X = 29.5\%$), Hg methylation rate constants ($k_m$) were remarkably well explained ($R^2Y = 95\%$) and predicted ($Q^2 = 78\%$) by the OPLS model I (Fig. 1c,d). This very high explanatory and predictive power demonstrates that Hg methylation rate constants can be quantitatively predicted from the molecular composition of sediment OM even in complex systems such as natural boreal lake sediments. The 10 compounds with the highest positive weight on the predictive component originate from fresh chlorophylls (phytene and phytol), proteins (aminopropanoyl leucine and 4 different diketodipiperazines) and phytoplankton-derived cell wall lipids ($n$-alkenes C20, C17 and C19) (Fig. 1c). In contrast, degraded OM and plant-derived compounds including fresh carbohydrates have either a negative weight on the predictive component or are part of the orthogonal component (O1: $R^2X = 25.8\%$; Fig. 1c). Our results therefore show that while plant-derived OM correlates negatively with Hg methylation, phytoplankton-derived OM compounds correlate strongly and positively with Hg methylation rate constants in boreal lake sediments.

Bacterial production rate (BP, µg C l$^{-1}$ d$^{-1}$) was significantly higher in lake sediments with OM of mainly autochthonous

origin (LOTS, M, VALE, V and STR, P value < 0.001). Compared with terrigenous OM, autochthonous carbon compounds are known to be preferentially used by heterotrophic bacteria[32]. Hence our results suggest that the OM molecular composition controls bacterial activity and thereby Hg methylation rates as both of these parameters are enhanced by phytoplankton-derived compounds. Indeed, a second OPLS model (model II, Supplementary Fig. 1) built with both $k_m$ and BP as Y variables showed that the abundance of phytoplankton-derived compounds (chlorophyll, protein, cell wall lipids) predicted the variability in both $k_m$ and BP (Supplementary Fig. 1). Furthermore, the second predictive component in this model suggested that among the phytoplankton-derived compounds, Hg methylation rate constants would be higher in the presence of fresh chlorophyll compounds and cell wall lipids, whereas BP was more strongly coupled to N-containing compounds derived from processed chlorophylls and proteins (that is, indole, methyl indole and maleimide) and from microbial chitin (that is, acetamidofuran and oxazoline). The OPLS model II shows therefore that the bacterial activity is well explained ($R^2Y = 96\%$) and can be well predicted ($Q^2 = 86\%$) by sediment OM molecular composition (Supplementary Fig. 1). In addition to the molecular composition of OM, a positive correlation between temperature of the water overlying the sediments and the presence of phytoplankton-derived OM (Fig. 3), and the sediment BP (P-value = 0.041), suggested a positive effect of temperature on phytoplankton-derived OM production and therefore on Hg methylation rates. Indeed, lakes dominated by phytoplankton-derived OM were warmer (Table 1) and shallower (Table 2) than those characterized by terrigenous OM inputs. The effect of environmental factors such as temperature, redox and concentrations of iron and sulfur on Hg methylation processes has been addressed previously[13,33,34]. For example, Korthals and Winfrey[33] showed that while temperature could explain 30% of the seasonal variation in Hg methylation in one specific lake, the variation in Hg methylation between sites was not significantly correlated to in situ temperature of different lakes. Drott et al.[13] showed that sulfide concentration can modulate the differences in $k_m$ at different sediment depths of an aquatic system, but the 'quality of OM' assessed by the C/N ratio defined the differences observed in $k_m$ between different sites. We thus conclude that OM molecular composition is the primary factor determining differences in BP and $k_m$ across boreal lakes.

Compared with the OPLS model I, BP as a single variable only explained 73% of the variation in Hg methylation rate (Fig. 2a) while 95% of the variance was explained by the OM molecular composition. This implies that besides the general stimulation of BP by phytoplankton-derived compounds, certain OM molecules are likely to specifically stimulate methylating bacterial populations and/or modulate Hg availability. There are currently no techniques for determining the concentration of Hg-complexes available for bacterial uptake in sediments. However the few studies that have assessed the role of different OM fractions on Hg availability for methylating bacteria in laboratory experiments[8,35–37] have highlighted a strong influence of low molecular mass thiols on bacterial Hg uptake[8,35]. Moreover, it has been suggested that algal and cyanobacterial exudates that contain low molecular mass thiols can enhance $k_m$ in periphytic biofilms[16]. The high $k_m$ observed in the sediments rich in phytoplankton-derived OM could be explained by a combined effect of enhanced activity of the microbial community, as bacterial communities preferentially use algal-derived OM for respiration[38], and a higher abundance of low molecular mass thiols derived from phytoplankton-derived exudates[16].

**Linking OM molecular composition with bulk parameters and Hg methylation.** Conventional bulk parameters such as THg,

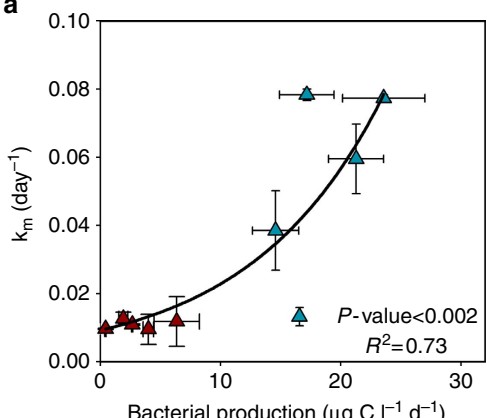

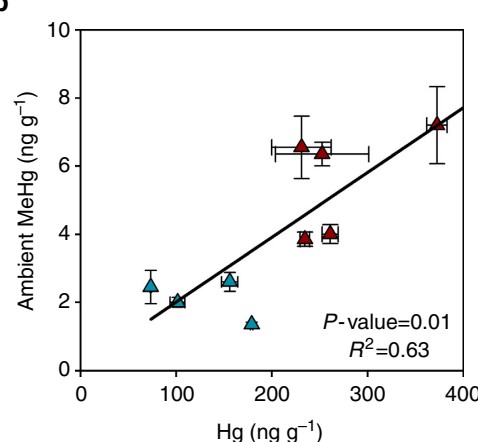

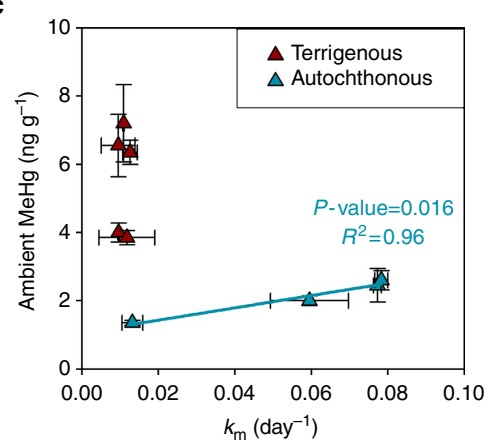

**Figure 2 | Relationships between Hg parameters and bacterial production for lakes dominated by autochthonous versus terrigenous OM.** Relationship between (**a**) Hg methylation rate constant ($k_m$ in day$^{-1}$) and bacterial production, (**b**) concentration of inorganic Hg (ng g$^{-1}$) and MeHg concentration (ng g$^{-1}$) and (**c**) Hg methylation rate constant and MeHg concentration (ng g$^{-1}$). The lake sediments dominated by terrigenous allochthonous OM (Lilla Sångaren, Ljustjärn, Svarttjärn, Fälaren, Oppsveten) are represented with red triangles. Lake sediments dominated by autochthonous OM (Strandsjön, Valloxen, Vallentunasjön, Marnästjärn and Lötsjön) are represented by green triangles. The lake Marnästjärn which is highly contaminated by historical anthropogenic inputs of Hg was not included in chart **c**. Error bars represent one standard deviation.

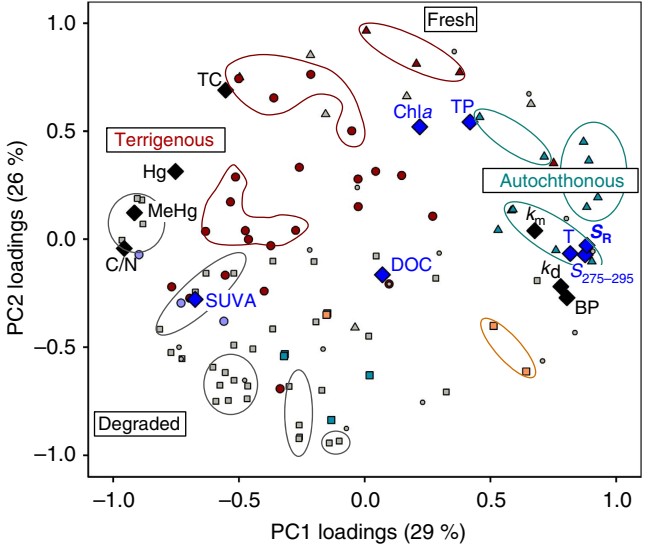

**Figure 3 | Comparison of bulk parameters and OM composition.**
Correlation analyses between conventional parameters (diamonds) measured in sediments (black) and their overlying water (blue) with the two first components of the PCA used to describe sediment OM molecular composition variation in the 10 studied boreal lakes. Similarly to Fig. 1, PC1/2-loadings of the pyrolytic organic compounds are sorted out into according to their origin (algal, plant, bacteria, invertebrate or unknown) and to their degradation status (fresh, degraded or unknown).

MeHg, DOC, TP, $O_2$, sulphate concentrations and optical properties were measured in a vertical profile of each water column (Table 2; Supplementary Table 1). Total C, TP, total N, C/N ratio and bacterial production rate (BP) were determined at two sediment depths (0–1 and 1–2 cm) for each lake (Supplementary Table 2). The comparison of sediment OM molecular composition with conventional OM parameters measured in sediments and in water overlying the sediment was carried out by reporting the correlation coefficients between these latter parameters and the scores of the two first principal components of the PCA (Fig. 3). The results confirm that high chlorophyll $a$ (chl $a$) and TP concentrations (common proxies for autochthonous production) in water overlying the sediment correlate with the presence of phytoplankton-derived OM in surface sediments. In contrast, $SUVA_{254}$, a traditional indicator of aromaticity[39], was positively correlated with (poly)aromatic pyrolytic compounds coming from terrestrial inputs (negative loadings on PC1). Therefore, conventional bulk OM parameters measured in water overlying the sediment reflected well the sediment OM molecular composition in terms of OM sources. In contrast, our results also indicate that DOC was not strongly coupled to any particular source of OM. The total C, Hg and MeHg in sediments were positively correlated with terrigenous OM compounds (negative loadings on PC1). The C/N ratio, previously suggested as a reliable proxy for the lability of OM modulating Hg methylation rates[13], was rather correlated with the presence of pyrolytic compounds derived from more extensively degraded terrigenous OM (negative loadings on PC1). While previous studies have reported low Hg methylation rates in sediments with high C/N ratio[13,14], our results demonstrate that lake sediments with low C/N ratios but that are enriched in chitin compounds derived from degraded planktonic OM (for example, STR) feature low Hg methylation rates (Fig. 1b). Hence the measurement of bulk parameters such as chl $a$ concentration or C/

N ratio can be useful to determine specific groups of organic compounds but fails at describing OM composition in the system. A detailed characterization of OM molecular composition is therefore required to disentangle the combined effects of different OM compounds on Hg methylation and to obtain precise and robust predictions of Hg methylation rates in boreal lake sediments.

**MeHg sources for boreal lakes.** To further describe the implications of OM molecular composition on MeHg cycling in boreal lakes, we investigated the relationship between *in situ* Hg methylation rate constants and sediment MeHg concentration in the two types of lake sediments, that is, sediments rich in autochthonous or in terrigenous OM. In lake sediments, the concentration of MeHg is determined by the Hg methylation rate ($k_m \times$ [Hg]) and MeHg demethylation rate ($k_d \times$ [MeHg]) and by MeHg import and export from the system[40]. While lake sediments dominated by terrigenous OM presented lower Hg methylation and MeHg demethylation rate constants than lakes enriched in phytoplankton-derived OM (Fig. 2b), they featured significantly higher Hg ($P$-value = 0.006) and MeHg concentrations ($P$-value = 0.004). High MeHg concentrations and decoupling of MeHg and either Hg methylation rate constants (Fig. 2c) or $k_m\,k_d^{-1}$ ratios (Supplementary Fig. 3a) in sediments dominated by terrigenous OM, suggest that MeHg levels in such sediments are most likely controlled by import of MeHg originally produced in the catchment and subsequently transported to lake sediments by surface runoff and other hydrological processes (Fig. 4b). Indeed, the correlations between terrestrially derived OM and concentrations of both Hg and MeHg (Fig. 3) points to catchment inputs of terrigenous Hg and MeHg. This is plausible considering that a large pool of Hg have accumulated in Swedish soils following atmospheric deposition during the industrial era[41] and that, recently, an increased released of DOC from soils to lakes has been linked to a two-fold increase in Hg in sediments[42]. By linking a detailed characterization of sediment OM molecular composition with Hg and MeHg transformation potentials, our study points to catchment inputs of terrigenous OM as the main source of Hg and MeHg for boreal lakes dominated by allochthonous OM.

In contrast, for lake sediments enriched in autochthonous phytoplankton-derived OM, the concentration of MeHg was positively correlated to the Hg methylation rate constant (Fig. 2c), the percentage of ambient MeHg (Supplementary Fig. 2b) and the $k_m\,k_d^{-1}$ ratio (Supplementary Fig. 3a). The latter two have often been used as proxies for net Hg methylation[43]. These correlations indicated that *in situ* Hg methylation determines MeHg levels in boreal lakes dominated by autochthonous phytoplankton-derived OM. Moreover, the OM molecular composition of lake sediments dominated by autochthonous OM showed that inputs from the boreal forest catchments were minor (Supplementary Table 3). With low Hg inputs from the catchment, atmospheric Hg deposition, which is similar for all the studied lakes[44], may thus be the primary source of Hg to lakes dominated by autochthonous OM (Fig. 4a). We conclude that processes determining MeHg concentrations in lakes dominated by phytoplankton-derived OM are different from those in lakes enriched in terrigenous OM (Fig. 4). While sediment MeHg in boreal lakes dominated by autochthonous OM inputs is mainly produced within the system, allochthonous MeHg formed in the surrounding catchment is likely the main source of MeHg for boreal lake sediments enriched in terrigenous OM.

**Discussion**
By using a sophisticated characterization of OM, we show for the first time that the molecular composition of OM is an important

Sources of MeHg for aquatic systems

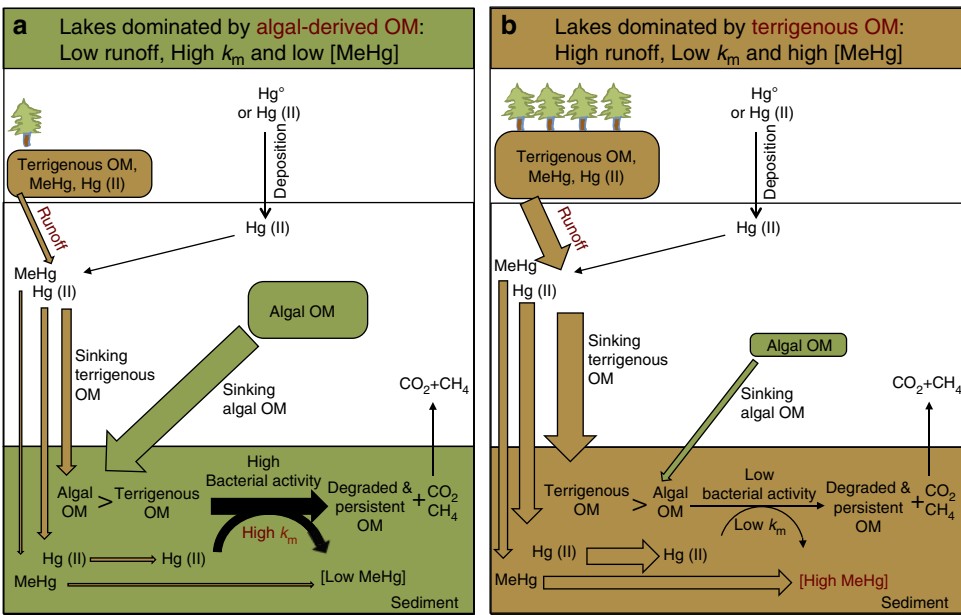

**Figure 4 | Conceptual model of MeHg sources for lake sediments.** (**a**) Lakes with high occurrence of planktonic blooms are enriched in fresh chlorophylls and proteins that enhance bacterial activity and MeHg formation; (**b**) increased runoff of terrigenous OM brings large amounts of Hg and MeHg but hampers *in situ* MeHg formation. MeHg in eutrophic lakes is the result of *in situ* production whereas runoff is the main source of MeHg for lakes dominated by terrigenous OM.

factor controlling the formation of neurotoxic MeHg in sediments and we provide insightful information to pinpoint the sources of Hg and MeHg in lake ecosystems. Based on our results and preceding work, we provide an updated framework for understanding and predicting MeHg concentration in boreal lake sediments (Fig. 4). We predict that enhanced phytoplankton blooms would likely cause increased Hg methylation rates and thus lead to higher MeHg levels in boreal lake sediments that receive low inputs of terrigenous OM (Fig. 4a). Also, increased import of terrigenous OM from surrounding catchments, for example, due to on-going climate change[45], may lead to higher MeHg concentrations in boreal lake sediments (Fig. 4b). In a global context of projected higher frequency of algal blooms caused by eutrophication[46] and increased global inputs of terrigenous organic matter to freshwater ecosystems[45], our findings bring new perspectives on how such future environmental changes may alter the biogeochemical cycling of Hg in boreal areas. Future studies are nevertheless needed to develop this framework further by including other environmental factors such as pH, redox and sulfur geochemistry known to modulate Hg availability and/or the activity of Hg methylating microorganisms; by considering also the bioaccumulation/biomagnification processes; and by extending it to other aquatic ecosystems such as alpine lakes, ponds, rivers, estuaries and sea.

## Methods

**Site selection and sampling.** We surveyed 10 lakes in central Sweden with different trophic status and receiving different amounts of terrigenous OM (Tables 1 and 2). We carried out two sampling campaigns: samples from M, and V, S, LS, O and F were retrieved in July 2012 and samples from LOTS, VALE, STR, LJU were collected in July 2013. Water column samples were collected with a GoFlo bottle (PVC). For Hg and MeHg measurements, 1 l of unfiltered water was placed in Teflon bottles and acidified (1% v/v final concentration; HCl Ultrex II, J. T. Baker). Within each lake, intact sediment cores with about 30 cm of overlying water were sampled with a 6-cm diameter gravity corer (UWITEC, Austria). After sampling, cores were kept upright at 4° C with

about 40 cm of overlying water until further processing within 12 h. All the steps involved in sediment handling and treatments were carried out in a $N_2$-filled glove box (Sigma-Aldrich, USA) to prevent oxidation of reduced chemical species. Sediment overlying water was first retrieved with acid-washed syringes from the sediment cores and then the upper 2 cm of the sediment core (0–1 and 1–2 cm) were sliced using acid washed plastic tools and used for further incubations or analyses. Sediment overlying waters were filtered through glass fiber filters (Whatmann, GF/F) and analysed for sulphate, DOC, TP, chlorophyll and optical OM properties (Supplementary Table 1). The remaining solid sediments were used for quantification of THg, MeHg and C, N, P concentrations, bacterial activity and Hg methylation ($k_m$) and demethylation ($k_d$) rates (Supplementary Table 2) and OM characterization (Supplementary Table 3).

**Water characteristics.** For optical characterization of OM in sediment overlying water, the absorbance spectra (200–800 nm) were measured with a Lambda 40 spectrophotometer (Perkin-Elmer, Waltham, USA). Spectral slope of absorbance coefficients between 275 and 295, and 350 and 400 nm, were obtained by non-linear fitting of the exponential model: $a_\lambda = a_{\lambda 0}\ e^{S(\lambda_0 - \lambda)}$ where $\lambda_0 > \lambda$ and $S$ is the spectral slope in the $\lambda_0 - \lambda$ spectral range[47–49]. The slope ratio, $S_R$, resulted from the ratio between $S_{275-295}$ nm and $S_{350-400}$ nm (ref. 48). We used the optical index ($SUVA_{254}$), which is related to the aromaticity, to compare our data with those from the literature[36]. The concentration of TP in the water was determined according to Murphy et al.[50] The DOC content from the water column was measured by high temperature catalytic oxidation (Shimadzu-TOC-L)[51]. $SUVA_{254}$ values were not corrected for iron concentrations.

**Sediment properties.** For measurements of total carbon and nitrogen concentrations about 7–10 mg of sediment were analysed by high-temperature catalytic oxidation with COTECH ECS 4010 elemental analyzer calibrated with sulfanilamide standard (C 41.84%, N 16.27%, H 4.68%, O 18.58%, S 18.62%). Analytical precision was < ± 0.4% for C and ± 2.1% for N. TP in sediments was measured as molybdate reactive phosphorus according to an established method[50]. Measurement of chlorophyll-*a* from phytoplankton was performed using ethanol as extraction solvent[52]. Bacterial production was measured after 1 h incubation at *in situ* temperature with 3H-labelled thymidine (Amersham, 1 mCi ml$^{-1}$, 80 Ci mmol$^{-1}$), at a final concentration of 12 nM (ref. 53). Leucine incorporation into protein was determined by precipitation with TCA and centrifugation, followed by scintillation counting with a Packard Tri-Carb 2100 TR liquid scintillation analyzer (PerkinElmer Life Sciences, Inc., Boston, USA). The leucine incorporation was converted into carbon units according to Simon and Azam[53].

**Hg analyses.** Potential rate constants for Hg methylation and MeHg demethylation were assayed by incubating the sliced sediments in sealed glass vials under dark and *in situ* temperature for 24 h with $^{198}Hg^{2+}$ and $^{204}Hg$-$CH_3$ isotope tracers added close to ambient concentrations. Hg species were extracted from the sediments by an acidic extraction assisted by microwave and then analysed by species specific isotope dilution and capillary gas chromatography hyphenated to inductively coupled plasma mass spectrometer (GC-ICPMS)[54].

**Molecular OM characterization.** Sedimentary OM composition was characterized using an oven pyrolyser equipped with an autosampler (PY-2020iD and AS-1020E, FrontierLabs, Japan) and connected to a GC/MS system (Agilent, 7890A-5975C, Agilent Technologies AB, Sweden), following the operating conditions optimized by Tolu et al.[20]. A data processing pipeline was employed to automatically integrate the peaks and extract the corresponding mass spectra under 'R' computational environment (version 2.15.2, 64 bits; http://www.R-projector.org). The compounds associated to the extracted mass spectra were identified under 'NIST MS Search 2' software using 'NIST/EPA/NIH 2011' library (http://chemdata.nist.gov/dokuwiki/doku.php?id=chemdata: ms-search) and additional published spectra. The 110 identified pyrolytic compounds are presented in Supplementary Table 5 with information about their origin and degradation status according to the literature review given in Tolu et al.[20]. For each sample, peak areas of the identified compounds were normalized to 100%.

**Statistical analyses.** Distributional properties of the data and transformations required to meet assumptions of normality (for example, $1/k_m$) were checked before the analysis of the data. The homogeneity in the variances was tested using the Levene test. Pearson correlations were used to test relationships between Hg methylation rate constants and conventional geochemical parameters. Regression analyses were carried out to evaluate relationships between BP, $k_m$, MeHg concentrations, %MeHg/THg and $k_m$ $k_d^{-1}$ (Sigmaplot 12). Univariate Analyses of Variance were applied to test for significant differences in the HgII, MeHg and $k_m$ between lake groups (autochthonous versus terrigenous) using SPSS software package PASW (version 22.0). PCA were performed with the 110 pyrolytic compounds and the 2 sediment depths for each lake also using SPSS 22.0. The principal components (PCs) were extracted using a Varimax rotated solution. The factor loading coefficients (PC-loadings) were calculated as regression coefficients (Fig. 1a,b). The multivariate calibration model between Hg methylation rate and bacterial production (variable $Y$) and the 110 pyrolytic organic compounds (variables $X$) was calculated and cross-validated using Orthogonal-Partial Least Square (O-PLS) regression in SIMCA software package, version 13 (Umetrics Umeå, Sweden). The quality of the model is described by $R^2Y$ and $Q^2$ values. $R^2Y$ represents the proportion of variance of the variable $Y$ (Hg methylation rate constants and bacterial production) explained by the model and indicates goodness of fit, while $Q^2$ corresponds the proportion of variance in the data predictable by the model and indicates predictability (Fig. 1c,d; Supplementary Fig. 1). Correlations between molecular composition of OM and conventional geochemical parameters (Fig. 3) were determined by principal components regression analysis.

**Data availability.** All data are presented in the paper as tables in the main manuscript or in the supporting information. Raw data will be provided upon request from the authors.

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

## Acknowledgements

We thank J Johansson and Claudia Bergin for assistance in the field and laboratory. Discussions with D Kothawala, S Sobek and B Obrador improved the manuscript. This research was funded by the Swedish Research Council (VR) grants to A.G.B. (project 2011–7192) and S.B. (grant 2012–3892 and 2013–6978), the Kempe Foundation (grants SMK-2745 to E.B. and SMK-2840 to Sy.B.) and the Swedish Research Council Formas (grant 2012–986 to S.B.). For J.T., the study has been conducted within a postdoctoral fellow financed by Umeå University and directed by Professor R Bindler and Professor J.-F. Boily.

## Author contributions

A.G.B., Sy.B., E.B. and St.B. conceived the study. A.G.B. conducted the two sampling campaigns with assistance from Sy.B. and A.M.R. A.G.B. and Sy.B. performed the GC–ICP–MS analyses of Hg with guidance from E.B. J.T. conducted Py-GCMS analyses and data treatment for the molecular characterization of OM. A.G.B. and J.T. conducted all statistical analyses and built the figures with comments and suggestions from Sy.B., St.B., E.B. and A.M.R. Finally, A.G.B, Sy.B and J.T. wrote the manuscript with significant assistance and comments from E.B., St.B. and A.M.R.

## Additional information

**Competing financial interests:** The authors declare no competing financial interests.

**Publisher's note**: 

