## [Peer Review File · Nature Communications]

Reviewers' Comments:

Reviewer #1 (Remarks to the Author)

This paper presents the results of a study designed to assess the influences of particulate organic matter in sediments of lakes located in Sweden on the microbial methylation of mercury. A major conclusion of the work is that 'fresh' autochthonous organic matter in the sediments fuels microbial activity and increases Hg methylation rates. The novelty of the research is not the observed influence of labile organic matter on increased rates of microbial activity, which has been recognized for some time, but the demonstration of this effect through the use of sophisticated characterization of the OM in sediments from lake systems influenced to different degrees by the presence of terrestrially derived particulate OM delivered from the lake watershed. Overall, I found the paper to be well written, informative and thought provoking. The most compelling data presented are those in Figure 1, especially fig. 1d. There are a few issues noted below that the authors should address:

Specific comments

1.) In general, it should be noted that pyrolysis is a non-specific compound, destructive method that doesn't necessarily provide direct info regarding compounds of interest. Rather the resulting compounds obtained in the analyses are indicative of the original compound types present in the sediment sample and they provide insights into the nature of the POM and its degree of degradation.

2.) It would strengthen the paper if porewater data, such as DOC (possibly further characterized by the methods used here), total filtered Hg, S=, and other potentially important constituents (e.g. Fe, NO₃⁻) were presented. The microbial world is primarily driven by dissolved constituents, not POC, where sorption to particles is considered a mode of preservation. Correlations of methylation across many systems are better predicted by dissolved constituents than those associated with the sediments. See for instance : Eagles-Smith, C.A., et al., Spatial and temporal patterns of mercury concentrations in freshwater fish across the Western United States and Canada, *Sci Total Environ* (2015), <http://dx.doi.org/10.1016/j.scitotenv.2016.03.229>.

The presentation of dissolved constituents in overlying waters is only partially useful as overlying waters do not necessarily reflect processes occurring in the sediments. No Hg or MeHg data are presented for overlying waters which are the pathways for food webs and potential export to down stream locations. Sediment bound Hg can be in many forms, most not bioavailable and potentially sequestered, along with sorbed MeHg, due to sorption. Dissolved Hg and MeHg are more likely to be transported, bioaccumulated, and photo-reacted (Hg reduction, MeHg demethylation).

Finally, I would recommend including NO₃⁻ data, as the presence of NO₃⁻ often inhibits MeHg formation by organisms lower on the redox ladder.

3.) P.5 and Methods. There is no indication when samples were collected from the lakes. It would be good to include data about the dates and status of the lakes at the time of sample collection - e.g. ice-out, spring flush inputs, before/after blooms, before/after turnover. Lakes are dynamic through the year, especially when dealing with the development of anoxic conditions required for methylation. These data would not change the results presented in Fig 1d as rate measurements were made on the collected sediments, but the data would provide important information regarding the conditions under which samples are collected and possibly provide more insights into the patterns observed with regard to MeHg in the sediments and could influence conclusions regarding watershed MeHg transported on particles to the lakes vs autochthonous production.

4.) P.5 and Table S1. Was Fe measured on the lake waters? High SUVA values (LS, S, O) may be influenced by the presence of Fe, as well as some intermediate values (STR). SUVA values greater than 5 are not common. SUVA data can be corrected for the presence of Fe. (see Poulin, B.A., Ryan, J.N., and Aiken, G.R., 2014, Effects of iron on optical properties of dissolved organic matter, Environmental Science and Technology, vol. 48, 10098-10106, dx.doi.org/10.1021/es502670r)

5.) P.7, lines 152-157. There are no data to support the suggestion that the Hg in the samples is associated with thiols in fresh OM, although one would expect Hg to be bound to thiols associated with OM in the absence of S=. The sediments are complicated chemical matrices and no info is presented regarding the presence of S=, a powerful Hg ligand (much stronger than thiols) in the sediments or pore fluids. This is also the case for other OM compounds (likely to contain thiols and other reduced S groups). I'd be cautious with this suggestion since just stimulating microbial activity is sufficient to drive increased methylation. Additional analyses are needed to address the issue of how the Hg is speciated to provide greater confidence to this supposition. In the next sentence (lines 158-159), for instance, the form of Hg is not listed as a factor in methylation in those systems with allochthonous sources, only the influences of labile materials. There could be substantial differences in the availability of Hg in autochthonous vs allochthonous systems resulting from differences in OM composition and Hg speciation, however, data to resolve this question are not presented in this paper.

6.) P. 8, Lines 192-194. The high concentrations of watershed derived Hg and MeHg referred to here are associated with sediment or particle bound Hg. A critical factor is the dissolved pool of Hg available for methylation. What is the primary source of Hg to the lakes showing greater methylation reactivity?

7.) P. 9. I think the concluding paragraph is too strongly worded and would suggest not overstating the significance of the results. The work has provided important insights into Hg methylation in the study lakes, but not all issues have been resolved.

First (lines 200-204), the study has not identified any specific compounds involved directly with the methylation of Hg. Rather they have traced the microbial degradation of particulate OM using a non-specific and destructive pyrolysis method. Given the difficulties of capturing rapidly cycling dissolved OM, I found it to be an elegant approach to assess OM lability in the particulate fractions and potentially follow the decreased lability of the OM pool over time, something that can be measured.

Second, (lines 206-207), the study focused on a handful of lakes in Sweden. Has the debate regarding Hg methylation in lakes been totally resolved? There are many other types of lake systems and many factors influencing Hg reactivity in lakes, so I'm not sure the issue has been universally resolved. The authors have more clearly demonstrated the need for labile OM to help drive microbial activity in the methylation process, but OM is just factor driving microbial activity.

Third, (lines 214-216), it isn't clear to me why the terrestrially derived MeHg, which has only been measured on particles in this study, would be more available to the food web than the MeHg formed in situ. What is unique regarding this material that it would preferentially be uptaken to a greater degree than the in situ MeHg? Wouldn't any MeHg in the system be potentially available? There is still much to be learned about bioaccumulation of Hg in lake food webs.

Reviewer #2 (Remarks to the Author)

This study tackles one of the most important question in the field of mercury biogeochemistry that is the nature of variables and overall mechanism controlling the availability of the potent neurotoxin monomethylmercury to food webs.

The role of natural organic matter on Hg cycling has received considerable attention but so far

provided contradictory results (promoting or hampering the toxin production) and prevented its proper integration in models aimed at predicting and managing mercury pollution.

This study is a timely contribution to the field. Its main strength is that it reconciles seemingly opposing views of the role of organic matter (OM) by offering its characterization at high resolution. This level of details in characterizing OM only became recently possible in sediments and the authors applied this knowledge to the field of Hg biogeochemistry.

The authors relied on multivariate statistical approaches to explore which of the OM properties were associated with MMHg production; they also developed a model with high explanatory and predictive power between the high methylation rate constants and the molecular composition of sedimentary OM.

The underlying mechanism emerging from their data is that the availability of specific organic compounds (of phytoplanktonic autochthonous origin) fuels bacterial productivity that in turn affects MMHg cycling (both production and destruction). It is particularly interesting that whereas bacterial productivity and k_m are stimulated by autochthonous OM, mercury methylation (k_m) appears to respond to specific compounds within the pool of autochthonous OM. This level of insight into the coupling of OM vs k_m , to the best of my knowledge, has not been possible thus far.

The exact nature of the molecules fuelling MMHg cycling remains unknown but the study clearly shows that it is derived from phytoplankton and that molecular-level characterization of the OM is required for precise and robust prediction of MMHg turnover in lake sediments.

Finally, the authors propose that in lakes dominated by algal derived OM, little MMHg will accumulate in food webs as the MMHg present is rapidly recycled resulting in a net low concentration of MMHg. On the contrary, in lakes dominated by terrigenous OM, MMHg is more likely to accumulate in food webs.

The study is sound and the manuscript is well written. Their data support the conclusions and speculation is kept to a necessary minimum.

One concern I have is that the link between this work and MMHg in food web is not properly addressed. This may seem anecdotal but in figure 3, the fate of sedimentary MMHg is not addressed and not linked (e.g., by an arrow) to "bioaccumulation". Many factors, some related to OM, will affect (even low levels of) MMHg transfer from sediment to top predators (affecting both bioaccumulation by organisms and bioamplification in food webs). Whereas this study is NOT about food web transfer but because of the potential broad impact of this work once published, it must be clarified that the MMHg issue does not stop at knowing what controls MMHg net production but that ecological process, also affected by OM, will control human and wildlife exposure to Hg. The ms alludes to food web processes on 215-216 but this needs more support in the discussion.

Details:

L96: What does it tell us that sediment OM composition is similar over 2 cm? I would have expected microbial processes and sedimentary diagenesis to affect OM quality and quantity with depth.

L104: Could the high invertebrate productivity in STR affect mercury methylation (lowering it) by oxygenating the sediment via bioturbation?

Figure 1b: it seems that the greatest rates of production (0.075-0.078) cluster together negatively on PC2 (autochthonous and degraded). The lowest methylation rates (M and STR) are at the extreme (very fresh or very degraded). If one were to draw a graph of $k_m = f(\text{degradation status})$, one would get a bell shaped curve, is that right? Would this data suggest that a mix of fresh and

degraded be necessary for optimal production of MMHg?

Figure 1c: Whereas the predictive component is associated with autochthonous vs. terrigenous (pink km square on the first axis), what do we know of the nature of the orthogonal components that affect MMHg production? Is this described on L130-133?

Figure S1, panel a: km is a purple square and BP is a red square. The caption mentions BP light green square and km, red square. Please clarify. What are the shaded areas?

Reviewer #3 (Remarks to the Author)

This manuscript investigates the role of organic matter quality in controlling methylmercury production in lake sediments. I congratulate the authors on their novel research. They have generated new information of broad interest for the field of mercury biogeochemistry, with implications for understanding the environmental fate of mercury pollution. The methods and study design are sound, although I found the data analysis and interpretation of results sometimes rather weak, and further clarification and revisions are recommended.

1) The sample size of 10 lakes is low, particularly in light of the broad conclusion made by the authors that their "findings reveal that the long debated controlling factor of methylmercury formation and accumulation in lake sediments is the origin of organic matter" (last line of abstract). The study sites only represent two lake types; eutrophic lakes with high inputs of autochthonous organic matter and brown water lakes with high loadings of terrestrial organic matter. There are many different types of lake ecosystems not represented in the study (e.g., alpine and arctic lakes with low terrestrial OM inputs and low primary production), and there exists a large gradient among lakes in the amount of terrestrial vs autochthonous contributions of OM. This limitation in the scope of the study needs to be acknowledged.

2) The linear regression analyses are questionable (e.g., figure 2) due to the issue of pseudoreplication and inflation of sample size. The replicate sediment layers (0-1 cm and 1-2 cm) were included as separate data points even though the sedimentary OM composition was very similar between layers (line 97, page 8). The inclusion of replicates as individual data points violates the assumption of independence of errors in the regression analysis. The mean values of those two layers should be analyzed instead. Given the lower sample size that would result from this revision (n=10 instead of 20), it is questionable whether the trends will still be statistically significant when lake-mean values are used. Even if so, a regression with 4 data points (e.g., Figure 2 C) would be rather weak evidence.

3) The bacterial production rates and mercury methylation rates were measured using in situ temperatures (lines 307, 316, page 15). There were much warmer in situ temperatures in the eutrophic lakes (11.5-18.8 {degree sign}C) compared to the lakes dominated by terrestrial OM inputs (4.8-18.7 {degree sign}C) with 4 lakes in the latter group having temperatures <10 {degree sign}C (Supplemental table 1). Therefore no standard temperature was used for rate measurements. How much of the variation in methylation rate or bacterial production is correlated to temperature? Is temperature more important than sediment OM composition in determining those rates?

4) It is unfortunate that methylmercury and total mercury concentrations were not measured in water above the sediment, at the same time that other water chemistry was measured (supplemental table 1). Those concentrations are important for demonstrating the overall impact of methylmercury production on water column (and food web) exposure to methylmercury.

5) The sediments dominated by terrestrial OM had lower methylation rates than those dominated by phytoplankton OM. The authors conclude from this result that plankton blooms stimulate mercury methylation and that sediments dominated by terrestrial OM have decreased bacterial

activity that hampers mercury methylation (Figure 3). An alternative explanation is that the inorganic mercury was less bioavailable for methylation in sediments with high terrestrial OM because the presence of more organic matter (or OM quality) resulted in greater binding efficiency. Could the results be an artifact of the method, where the injected spikes of inorganic mercury were less bioavailable in the sediments dominated by terrestrial OM? This is an important point because a key conclusion of the study is that the lability of the OM for bacterial degradation is what controls the mercury methylation rate.

6) Supplemental table 2 shows that the %methylmercury contents are comparable in sediments dominated by both types of OM. Other studies have demonstrated that the %methylmercury content of sediments is a useful proxy for mercury methylation rate. Why is there a discrepancy in this dataset, where the sediments dominated by terrestrial OM have high methylmercury concentrations, high %methylmercury content but low methylation rates? The explanation provided is that the methylmercury in those sediments originates primarily from terrestrial/catchment sources. What evidence is available to support this assumption? Where measurements made of catchment loadings of methylmercury to those lakes? Hg loading can vary widely depending on catchment characteristics, hydrology and lake morphometry. More information is warranted to support their explanation.

7) The discussion and analysis of dominant controls on mercury methylation in sediments is overly simplistic because there is little consideration of other environmental factors that have been previously demonstrated to play a key role such as pH, redox conditions, and temperature. The potential influences of those other factors need to be acknowledged, in the very least in the discussion and even better in the statistical analysis.

8) Lines 214-216, page 9. The statement that methylmercury transported from terrestrial OM is more likely to be accumulate in the aquatic food chain is not broadly accepted in the field, and other studies have concluded that in situ methylmercury production is a greater source to the water column than catchment inputs (e.g., Watras et al. Environ Sci Technol. 2005 Jul 1;39(13):4747-58). The statement is misleading.

9) There is a contradiction in the emphasis and conclusions of the study. Lakes with sediment dominated by terrestrial OM showed lower mercury methylation rates but higher concentrations of methylmercury than sediment dominated with plankton OM, where higher mercury methylation rates were found. If this difference is explained by catchment loadings of methylmercury, then isn't the dominant factor controlling levels of methylmercury in sediments of the study lakes more related to the extent of catchment loading rather than the quality of the organic matter? Although mercury methylation rates were stimulated by planktonic blooms, the effect was less than that of external loading of methylmercury. How relevant than is the quality of OM for controlling methylmercury accumulation in sediments?

Reviewer #1 (Remarks to the Author):

This paper presents the results of a study designed to assess the influences of particulate organic matter in sediments of lakes located in Sweden on the microbial methylation of mercury. A major conclusion of the work is that 'fresh' autochthonous organic matter in the sediments fuels microbial activity and increases Hg methylation rates. The novelty of the research is not the observed influence of labile organic matter on increased rates of microbial activity, which has been recognized for some time, but the demonstration of this effect through the use of sophisticated characterization of the OM in sediments from lake systems influenced to different degrees by the presence of terrestrially derived particulate OM delivered from the lake watershed. Overall, I found the paper to be well written, informative and thought provoking. The most compelling data presented are those in Figure 1, especially fig. 1d. There are a few issues noted below that the authors should address:

Specific comments

1.) In general, it should be noted that pyrolysis is a non-specific compound, destructive method that doesn't necessarily provide direct info regarding compounds of interest. Rather the resulting compounds obtained in the analyses are indicative of the original compound types present in the sediment sample and they provide insights into the nature of the POM and its degree of degradation.

We acknowledge that pyrolysis-GC-MS is a destructive method compared to high-throughput spectroscopic methods or those based on wet chemical extractions and LC/GC-MS. However, it provides specific pyrolytic compounds for several different types/classes of organic molecules (e.g., lignin, chlorophyll, proteins, acetamino-sugars) deriving from different sources and known to have a different microbial and physicochemical reactivity in the environment. We have included a discussion about the advantages and limitations of Py-GC-MS method in comparison to others existing methods to characterize OM, see lines 65–74. Also, we have now provided a more extensive description about the compounds we have identified and the molecules they derive from, with an emphasis on their origin and degree of degradation. Please see lines 76–99 of the MS.

2.) It would strengthen the paper if porewater data, such as DOC (possibly further characterized by the methods used here), total filtered Hg, S₂, and other potentially important constituents (e.g Fe, NO₃⁻) were presented. The microbial world is primarily driven by dissolved constituents, not POC, where sorption to particles is considered a mode of preservation. Correlations of methylation across many systems are better predicted by dissolved constituents than those associated with the sediments. See for instance : Eagles-Smith, C.A., et al., Spatial and temporal patterns of mercury concentrations in freshwater fish across

the Western United States and Canada, *Sci Total Environ* (2015),
<http://dx.doi.org/10.1016/j.scitotenv.2016.03.229>.

The presentation of dissolved constituents in overlying waters is only partially useful as overlying waters do not necessarily reflect processes occurring in the sediments. No Hg or MeHg data are presented for overlying waters which are the pathways for food webs and potential export to down stream locations. Sediment bound Hg can be in many forms, most not bioavailable and potentially sequestered, along with sorbed MeHg, due to sorption. Dissolved Hg and MeHg are more likely to be transported, bioaccumulated, and photo-reacted (Hg reduction, MeHg demethylation). Finally, I would recommend including NO₃- data, as the presence of NO₃- often inhibits MeHg formation by organisms lower on the redox ladder.

We agree with the reviewer that a combined assessment of the influence of both porewater chemistry and sediment OM composition on Hg methylation rates and Hg speciation would have been useful. However the volumes of sediments and porewaters available before/after incubations were not large enough to measure all these parameters and we have been restricted in that way. Moreover, attempts to correlate dissolved compounds (IHg, H₂S, Fe, etc) with methylation rates have been explored in previous studies¹⁻⁴. We therefore focused our efforts on verifying our hypothesis, that sedimentary OM molecular composition (i.e. with molecular data on OM composition and not only bulk OM parameters as previously done) controls Hg methylation rate constants in boreal lake sediments. We believe this objective is better addressed with a molecular analysis of the solid OM rather than DOC characterization because, as the reviewer mentioned, sorption to POC is considered a preservation mode for OM. We have now carried out a correlation analyses (Fig. 3) to show the correspondence between some bulk parameters, such as chl *a* concentrations, $S_{275-295}$ and $SUVA_{254}$ measured in the water overlying the sediment and with specific groups of pyrolytic OM compounds detected by Py-GC-MS in surface sediments. The new Figure 3 shows that Chl *a* and TP concentrations and DOM optical properties measured in overlying waters are in good qualitative agreement with the molecular OM composition of the sediments. Please see the discussion in lines 196–227. Unfortunately we did not measure NO₃ concentrations in these lakes. Furthermore, we have added a vertical profile of IHg and MeHg concentrations and geochemical characteristics of the studied lakes, see table 2.

MOLECULAR COMPOSITION OF SEDIMENTARY ORGANIC MATTER
(PRINCIPAL COMPONENT ANALYSIS) AND BULK PARAMETERS

3.) P.5 and Methods. There is no indication when samples were collected from the lakes. It would be good to include data about the dates and status of the lakes at the time of sample collection -e.g. ice-out, spring flush inputs, before/after blooms, before/after turnover. Lakes are dynamic through the year, especially when dealing with the development of anoxic conditions required for methylation. These data would not change the results presented in Fig 1d as rate measurements were made on the collected sediments, but the data would provide important information regarding the conditions under which samples are collected and possibly provide more insights into the patterns observed with regard to MeHg in the sediments and could influence conclusions regarding watershed MeHg transported on particles to the lakes vs autochthonous production.

We agree with the reviewer and we have added a more detailed description about the sampling period and lake geochemical characteristics has been added in the revised manuscript. Please see new tables 1 and 2 and lines 57–62 and lines 281–287.

4.) P.5 and Table S1. Was Fe measured on the lake waters? High SUVA values (LS, S, O) may be influenced by the presence of Fe, as well as some intermediate values (STR). SUVA values greater than 5 are not common. SUVA data can be corrected for the presence of Fe. (see Poulin, B.A., Ryan, J.N., and

Aiken, G.R., 2014, Effects of iron on optical properties of dissolved organic matter, *Environmental Science and Technology*, vol. 48, 10098-10106, [dx.doi.org/10.1021/es502670r](https://doi.org/10.1021/es502670r)).

We agree with the reviewer that iron might influence SUVA₂₅₄ values in enriched iron environments. We did not measure Fe concentrations in all of the lakes, but did so for LJU (SUVA=1,2), LS (SUVA=3,3) and S (SUVA=5). In general the studied lakes are poor in iron. Indeed for LJU and LS, the Fe concentrations were below the analytical detection limit (85 µg L⁻¹). S in contrast presented the highest measured iron concentration amounting to 0.5 mg L⁻¹. As we do not have iron concentration for all the studied systems, and it seems appropriate to treat all the samples in the same way, we decided to not correct any SUVA values for Fe but instead include this point in the methods (line 309). Either way, the new Fig 3 confirms that despite potential iron effects, the measured SUVA values in the studied lakes are highly correlated with poly-aromatic compounds identified by Py-GC-MS.

5.) P.7, lines 152-157. There are no data to support the suggestion that the Hg in the samples is associated with thiols in fresh OM, although one would expect Hg to be bound to thiols associated with OM in the absence of S=. The sediments are complicated chemical matrices and no info is presented regarding the presence of S=, a powerful Hg ligand (much stronger than thiols) in the sediments or pore fluids. This is also the case for other OM compounds (likely to contain thiols and other reduced S groups). I'd be cautious with this suggestion since just stimulating microbial activity is sufficient to drive increased methylation. Additional analyses are needed to address the issue of how the Hg is speciated to provide greater confidence to this supposition. In the next sentence (lines 158-159), for instance, the form of Hg is not listed as a factor in methylation in those systems with allochthonous sources, only the influences of labile materials. There could be substantial differences in the availability of Hg in autochthonous vs allochthonous systems resulting from differences in OM composition and Hg speciation, however, data to resolve this question are not presented in this paper.

We have no data to directly support the suggestion that the Hg in the samples is associated with thiols in fresh OM. However the fact that BP alone explained only 73% of the measured variation in Hg methylation rate compared to 95% for OM composition (Figure 2a), other factors beyond a general stimulation of BP by phytoplankton-derived compounds are likely to influence this process. As discussed by the reviewer, differences in Hg speciation and bioavailability have already been evidenced for different OM sources/reactivities and we now included this discussion in lines 183–195.

6.) P. 8, Lines 192-194. The high concentrations of watershed derived Hg and MeHg referred to here are associated with sediment or particle bound Hg. A

critical factor is the dissolved pool of Hg available for methylation. What is the primary source of Hg to the lakes showing greater methylation reactivity?

An increase on Hg burial in sediments of boreal lakes has been recently related to an increase of terrigenous C⁵. Our results also indicate that lakes dominated by terrigenous OM (low km) received more inputs of Hg from the catchment (Fig 2). Also a correlation analyses between OM molecular composition and Hg and MeHg concentrations (Fig. 3) suggest that inputs of terrigenous OM from the surrounding catchment is the main source of Hg for the studied boreal lake ecosystems. We did not measure the fluxes of Hg entering the studied systems but it is plausible that in lakes dominated by autochthonous OM (high km), located far from Hg point sources and with lower inputs of terrigenous OM, atmospheric Hg deposition may actually also be an important source of Hg to the system. We have now included this cautionary comment in the discussion about Hg methylation processes (lines 247–255) and also in the response to the last comment of the reviewer 3.

7.) P. 9. I think the concluding paragraph is too strongly worded and would suggest not overstating the significance of the results. The work has provided important insights into Hg methylation in the study lakes, but not all issues have been resolved.

First (lines 200-204), the study has not identified any specific compounds involved directly with the methylation of Hg. Rather they have traced the microbial degradation of particulate OM using a non-specific and destructive pyrolysis method. Given the difficulties of capturing rapidly cycling dissolved OM, I found it to be an elegant approach to assess OM lability in the particulate fractions and potentially follow the decreased lability of the OM pool over time, something that can be measured.

A better description about the method is now provided in lines 65–74 and 76–99. Also, according to reviewer's comment we have changed the abstract and the last paragraph to avoid overstating the significance of the results. Lines 20–22 and the concluding paragraph lines 274–278.

Second, (lines 206-207), the study focused on a handful of lakes in Sweden. Has the debate regarding Hg methylation in lakes been totally resolved? There are many other types of lake systems and many factors influencing Hg reactivity in lakes, so I'm not sure the issue has been universally resolved. The authors have more clearly demonstrated the need for labile OM to help drive microbial activity in the methylation process, but OM is just factor driving microbial activity.

We agree with the reviewer. We have now more clearly emphasized the focus on lakes in the boreal zone (see lines 51–54) and concluded that similar approaches should be incorporated in future studies of Hg methylation processes in other types of waters, such as alpine lakes, arctic areas and aquatic systems ponds,

rivers, estuaries and sea. Please see lines 274–278.

Third, (lines 214-216), it isn't clear to me why the terrestrially derived MeHg, which has only been measured on particles in this study, would be more available to the food web than the MeHg formed in situ. What is unique regarding this material that it would preferentially be uptaken to a greater degree than the in situ MeHg? Wouldn't any MeHg in the system be potentially available? There is still much to be learned about bioaccumulation of Hg in lake food webs.

We agree with the reviewer that the effects of mercury speciation and origin of OM on Hg bioaccumulation is still not fully resolved and that further studies are needed. As our study did not explore this aspect of the mercury cycle, and based on the comments from the three reviewers, we have therefore decided to remove this rather speculative paragraph from the discussion and simply emphasize the need for further studies on this topic (line 278).

Reviewer #2 (Remarks to the Author):

This study tackles one of the most important question in the field of mercury biogeochemistry that is the nature of variables and overall mechanism controlling the availability of the potent neurotoxin monomethylmercury to food webs.

The role of natural organic matter on Hg cycling has received considerable attention but so far provided contradictory results (promoting or hampering the toxin production) and prevented its proper integration in models aimed at predicting and managing mercury pollution.

This study is a timely contribution to the field. Its main strength is that it reconciles seemingly opposing views of the role of organic matter (OM) by offering its characterization at high resolution. This level of details in characterizing OM only became recently possible in sediments and the authors applied this knowledge to the field of Hg biogeochemistry.

The authors relied on multivariate statistical approaches to explore which of the OM properties were associated with MMHg production; they also developed a model with high explanatory and predictive power between the high methylation rate constants and the molecular composition of sedimentary OM.

The underlying mechanism emerging from their data is that the availability of specific organic compounds (of phytoplanktonic autochthonous origin) fuels bacterial productivity that in turn affects MMHg cycling (both production and destruction). It is particularly interesting that whereas bacterial productivity and km are stimulated by autochthonous OM, mercury methylation (km) appears to respond to specific compounds within the pool of autochthonous OM. This level of insight into the coupling of OM vs km, to the best of my knowledge, has not been possible thus far. The exact nature of the molecules fuelling MMHg cycling remains unknown but the study clearly shows that it is derived from phytoplankton and that molecular-level characterization of the OM is required for

precise and robust prediction of MMHg turnover in lake sediments.

Finally, the author propose that in lakes dominated by algal derived OM, little MMHg will accumulate in food webs as the MMHg present is rapidly recycled resulting in a net low concentration of MMHg. On the contrary, in lakes dominated by terrigenous OM, MMHg is more likely to accumulate in food webs.

The study is sound and the manuscript is well written. Their data support the conclusions and speculation is kept to a necessary minimum.

One concern I have is that the link between this work and MMHg in food web is not properly addressed. This may seem anecdotal but in figure 3, the fate of sedimentary MMHg is not addressed and not linked (e.g., by an arrow) to "bioaccumulation". Many factors, some related to OM, will affect (even low levels of) MMHg transfer from sediment to top predators (affecting both bioaccumulation by organisms and bioamplification in food webs). Whereas this study is NOT about food web transfer but because of the potential broad impact of this work once published, it must be clarified that the MMHg issue does not stop at knowing what controls MMHg net production but that ecological process, also affected by OM, will control human and wildlife exposure to Hg. The ms alludes to food web processes on 215-216 but this needs more support in the discussion.

The former Figure 3, now Figure 4 has been modified. Based on reviewer's comments, we have now modified the figure to exclude any inferred MeHg bioaccumulation or biomagnification as this aspect of the Hg cycle was not directly addressed in our experimental work and would thus be speculative. We now simply state in the text that more work is needed in this area (line 278)

New Fig. 4

Details:

L96: What does it tell us that sediment OM composition is similar over 2 cm? I would have expected microbial processes and sedimentary diagenesis to affect OM quality and quantity with depth.

With the same assumption mentioned by the reviewer, we first investigated the two sediment depths (0-1 and 1-2 cm) in order to assess how differences in OM composition over depth coupled with degradation of the organic matter influenced Hg methylation rates. We expected a higher degree of degradation with increasing depth, but our data did not reveal any such trends in the studied lakes. The differences in the OM composition between the two sediments depths are exemplified in the table below with the proportions of levosugars (fresh carbohydrates), proteins, and phytol (fresh chlorophyll), which are known pyrolytic products of organic compounds that are rapidly assimilated and degraded⁶⁻⁸. For some compounds and/or lake, we do see a clear decrease of their proportions with sediment depth (highlighted in orange in the table below), but in most of the cases the decrease in their proportions are within 5-10% (i.e. within the analytical and/or sampling uncertainty; highlighted in green). Moreover, we also observed an increase in the proportion of these labile compounds over sediment depth for some lakes (highlighted in blue). The fact that we could not observe clear OM degradation with sediment depth cannot be ascribed to the analytical method used for characterizing OM. Indeed, when applied to a varved sediment core, this method has clearly shown OM degradation with increased sediment depth over a period of 15 year which was associated with a clear loss in levosugars, proteins and phytol⁸. In the studied lakes, the patterns for OM degradation with sediment depth is most probably attenuated by surface sediment mixing from e.g. bioturbation which is absent from such varved sediments.

We nevertheless acknowledge the reviewer's comment and to address this we have added a table to the SI (Table S4) and inserted some new text on lines 116-122 of the MS.

Table S4. Differences in TC and TN contents as well as in the relative abundances of proteins, levosugars (fresh carbohydrates) and phytol (fresh chlorophyll) between the sediments at 0-1 and 1-2 cm. Decreases on the relative abundances of specific organic compounds with depth are highlighted in orange. Decreases in the relative abundances of the organic compounds that fall into 5-10% (analytical and/or sampling uncertainty) are highlighted in green. Increase in the relative abundance of these labile organic compound with sediment depth for some lakes are highlighted in blue.

	TC (%)		TN (%)		Levosugar		Proteins		Phytol	
	Individual values	Av ± sd (rsd)	Individual values	Av ± sd (rsd)	Individual values	Av ± sd (rsd)	Individual values	Av ± sd (rsd)	Individual values	Av ± sd (rsd)
F 0-1	24.6	24.4 ± 0.3	1.96	1.95 ± 0.01	4.0	3.9 ± 0.1	3.25	3.23 ± 0.03	0.21	0.18 ± 0.04
F 1-2	24.2	(1%)	1.94	(1%)	3.8	(4%)	3.20	(1%)	0.16	(21%)
LJU 0-1	31.77	31.9 ± 0.2	2.48	2.47 ± 0.02	8.5	8.54 ± 0.05	3.02	2.96 ± 0.08	0.11	0.10 ± 0.01
LJU 1-2	32.04	(1%)	2.45	(1%)	8.6	(1%)	2.91	(6%)	0.09	(12%)
LOTS0-1	13.91	14.3 ± 0.6	1.75	1.75 ± 0.01	5.1	4.9 ± 0.3	4.83	4.7 ± 0.1	0.95	0.89 ± 0.09
LOTS 1-2	14.76	(4%)	1.74	(0.4%)	4.7	(8%)	4.64	(3%)	0.83	(10%)
LS 0-1	19.8	19 ± 1	1.34	1.3 ± 0.1	0.6	1.4 ± 1.2	2.41	2.3 ± 0.2	0.09	0.8 ± 0.1
LS 0-2	18.4	(5%)	1.20	(8%)	2.3	(86%)	2.18	(7%)	0.08	(12%)
M 0-1	19.97	19.7 ± 0.3	2.21	2.18 ± 0.05	7.5	7.4 ± 0.1	5.11	5.0 ± 0.2	2.05	2.1 ± 0.1 (6%)
M 1-2	19.48	(2%)	2.14	(2%)	7.3	(2%)	4.89	(3%)	2.22	
O 0-1	19.0	19.2 ± 0.3	1.10	1.10 ± 0.01	0.4	0.6 ± 0.4	2.02	1.99 ± 0.03	0.04	0.05 ± 0.01
O 1-2	19.4	(2%)	1.10	(1%)	0.9	(56%)	1.97	(1%)	0.06	(22%)
S 0-1	24.3	23.7 ± 0.8	1.54	1.49 ± 0.08	11.2	10 ± 2	3.35	3.3 ± 0.1	0.09	0.08 ± 0.01
S 1-2	23.1	(4%)	1.43	(5%)	8.2	(26%)	3.19	(3%)	0.08	(10%)
STR 0-1	11.66	11.7 ± 0.1	1.35	1.35 ± 0.01	1.0	0.8 ± 0.2	4.34	4.3 ± 0.06	0.05	0.07 ± 0.02
STR 1-2	11.81	(1%)	1.34	(0.05%)	0.6	(31%)	4.26	(2%)	0.08	(31%)
V 0-1	14.0	13.9 ± 0.2	1.72	1.72 ± 0.01	3.6	2.4 ± 1.7	5.58	5.4 ± 0.2	4.61	4.8 ± 0.3 (5%)
V 1-2	13.7	(2%)	1.71	(0.4%)	1.2	(72%)	5.25	(5%)	4.97	
VALE 0-1	18.69	18.9 ± 0.2	2.37	2.38 ± 0.03	13.7	13 ± 2	5.84	5.84 ± 0.01	2.49	2.8 ± 0.4
VALE 1-2	19.04	(1%)	2.38	(0.3%)	11.6	(12%)	5.83	(0.2%)	3.04	(14%)

L104: Could the high invertebrate productivity in STR affect mercury methylation (lowering it) by oxygenating the sediment via bioturbation?

We actually discarded this possibility as the water overlying the sediment at STR was anoxic. We now provide a more extensive geochemical characterization of the vertical profiles of the water column in Table 2 of the MS.

Figure 1b: it seems that the greatest rates of production (0.075-0.078) cluster together negatively on PC2 (autochthonous and degraded). The lowest methylation rates (M and STR) are at the extreme (very fresh or very degraded). If one were to draw a graph of $k_m=f(\text{degradation status})$, one would get a bell shaped curve, is that right? Would this data suggest that a mix of fresh and degraded be necessary for optimal production of MMHg?

The reviewer raises an interesting point. We would like to remind that the PCA analysis were carried out uniquely with the identified organic compounds, and aims at understanding/showing how the OM molecular composition varies in the studied lakes. Thus, we cannot use the PCA to conclude about the compounds correlating with k_m . We used the The OPLS model I to determine the organic compounds that correlate with k_m . The OPLS model I shows that fresh algal derived compounds (proteins, phytol) have the highest positive weights (i.e. strongly correlate positively to k_m) on the predictive component (Fig 1c). We however only found one organic compound resulting from degradation processes that positively correlated with k_m (Fig. 1c). Thus, with our results we cannot conclude that a mix of fresh and degraded OM is necessary for the optimal production of MeHg.

Figure 1c: Whereas the predictive component is associated with autochthonous vs. terrigenous (pink k_m square on the first axis), what do we know of the nature of the orthogonal components that affect MMHg production? Is this described on L130-133?

We have now included a better description on the significance of the two types (predictive versus orthogonal) components of the OPLS model. Please see lines 131–138, 145–150.

Figure S1, panel a: k_m is a purple square and BP is a red square. The caption mentions BP light green square and k_m , red square. Please clarify. What are the shaded areas?

We apologize and we thank the reviewer for finding the mistake. We have now corrected it and BP is now described as a red square and k_m as a pink square in the caption.

Reviewer #3 (Remarks to the Author):

This manuscript investigates the role of organic matter quality in controlling methylmercury production in lake sediments. I congratulate the authors on their novel research. They have generated new information of broad interest for the field of mercury biogeochemistry, with implications for understanding the environmental fate of mercury pollution. The methods and study design are sound, although I found the data analysis and interpretation of results sometimes rather weak, and further clarification and revisions are recommended.

1) The sample size of 10 lakes is low, particularly in light of the broad conclusion made by the authors that their "findings reveal that the long debated controlling factor of methylmercury formation and accumulation in lake sediments is the origin of organic matter" (last line of abstract). The study sites only represent two lake types; eutrophic lakes with high inputs of autochthonous organic matter and brown water lakes with high loadings of terrestrial organic matter. There are many different types of lake ecosystems not represented in the study (e.g., alpine and arctic lakes with low terrestrial OM inputs and low primary production), and there exists a large gradient among lakes in the amount of terrestrial vs autochthonous contributions of OM. This limitation in the scope of the study needs to be acknowledged.

We agree with the reviewer that our study only covers the boreal area and there are still many biomes where the effect of OM molecular composition on Hg methylation processes remains to be studied. At a global scale, boreal biomes are very important in hosting 28.4% of the global lake area (derived from dividing the total estimated boreal lake area⁹ by the global lake area¹⁰). Within the boreal area, our ten lakes cover a wide range of trophic states where sediments receive different amounts of terrigenous and/or planktonic derived OM. We have now emphasized this point on lines 278. We have replaced the previous last sentence of the abstract with a new sentence, lines 20–22. Furthermore, we have highlighted throughout the MS our focus on the boreal ecosystem (including the title and abstract).

2) The linear regression analyses are questionable (e.g., figure 2) due to the issue of pseudoreplication and inflation of sample size. The replicate sediment layers (0-1 cm and 1-2 cm) were included as separate data points even though the sedimentary OM composition was very similar between layers (line 97, page 8). The inclusion of replicates as individual data points violates the assumption of independence of errors in the regression analysis. The mean values of those two layers should be analyzed instead. Given the lower sample size that would result from this revision (n=10 instead of 20), it is questionable whether the trends will still be statistically significant when lake-mean values are used. Even if so, a

regression with 4 data points (e.g., Figure 2 C) would be rather weak evidence.

The Figure 2 and Supplementary Fig. 2 and Supplementary Fig. 3 have been changed according to the reviewer's comment. Statistical regression analyses have been carried out and reported in the figures. For figure 1 we however keep the two sediment depth layers from individual lakes separate for two main reasons. First, the PCA and OPLS model 1 carried out with the two different sediment depths provide insightful information on the effect of the degradation status of OM on k_m with sediment depth (please see also our response to comments from reviewer 2 on line 96 (see above) and the new supplementary table S4). Second, the OPLS statistical model carried out with only the upper sediment layer provides the same information and conclusions as the model presented in the MS (see below).

When the OPLS model I (Y variable= k_m ; X variables=organic compounds) is carried out with the mean of the two sediment layers of each lake (thus, $n=10$), exactly the same distribution of the organic compounds on the predictive and orthogonal components is obtained as with the OPLS model I presented in the manuscript based on the two sediment depths (cf. figure below versus Fig 1c in the manuscript). The output and the associated conclusions of both OPLS model 1 in the MS and the OPLS model presented below point to fresh phytoplankton-derived OM compounds correlating positively with k_m . Moreover, the OPLS model based on the mean only explains a bit less of variability in k_m ($R^2Y=86.6\%$ versus 94.5%) and has only a slightly lower power of prediction ($Q = 71$ versus 78%) than the model I based on the 20 samples as presented in the manuscript

OPLS statistical model carried out with the mean of the two sediment layers of each lake (thus, $n=10$).

3) The bacterial production rates and mercury methylation rates were measured using in situ temperatures (lines 307, 316, page 15). There were much warmer in situ temperatures in the eutrophic lakes (11.5-18.8 {degree sign}C) compared to the lakes dominated by terrestrial OM inputs (4.8-18.7 {degree sign}C) with 4 lakes in the latter group having temperatures <10 {degree sign}C (Supplemental table 1). Therefore no standard temperature was used for rate measurements. How much of the variation in methylation rate or bacterial production is correlated to temperature? Is temperature more important than sediment OM composition in determining those rates?

This is an interesting point. Korthals and Winfrey¹¹ showed that the variation in Hg methylation rates between sites was not significantly correlated ($r^2 < 0.093$) to in situ temperature of different lakes. However, they observed that within a system (Lake Clara) changes in temperature could explain 30% of the variation on Hg methylation. To evaluate the potential importance of T to our results we have compared two models: i) OPLS built with the OM molecular composition of the surface sediment ii) OPLS model with the OM molecular composition of the surface sediment and the T measured in water overlying the sediment. The explained variability and power of prediction in k_m of the OPLS model built uniquely with molecular OM composition (R²Y= 86.6%; Q=71.2%) and the OPLS model built with OM plus T (R²Y= 86.5%; Q=70.8%) are very similar. Thus compared to OM molecular composition, T does not explain more of the observed variability. In the MS, the OPLS model II actually includes BP (two Y). This model indicates that OM molecular composition has a strong effect not only on k_m but also on the activity of bacteria. To respond to this comment we have also built an OPLS model that predicts exclusively BP (R²Y= 97.5%; Q=84.5%). The model shows that the molecular composition of OM explains 97.5% of the variability in the BP. It has been shown that the persistence or reactivity of OM in lakes depend on its molecular characteristics¹². Accordingly, once terrestrial OM is mobilized and diluted into aquatic environments, the major controls of degradation switch from extrinsic (environmental factors) to intrinsic (molecular composition)¹². In line with this observation, our study shows that differences on OM molecular composition across systems explain more of the variability in BP and k_m than do differences in T. However, previous studies¹¹ show that changes in T will affect k_m within a system. We thank the reviewer for this comment (and comment 7), together they have helped us to disentangle the role of intrinsic (OM molecular composition) and extrinsic factors (such as T) on Hg methylation processes within a lake and across different lakes. This has been now addressed in lines 163–178.

4) It is unfortunate that methylmercury and total mercury concentrations were not measured in water above the sediment, at the same time that other water chemistry was measured (supplemental table 1). Those concentrations are

important for demonstrating the overall impact of methylmercury production on water column (and food web) exposure to methylmercury.

For the lakes sampled in 2012, we did measure these concentrations. We now include this data in table 2.

5) The sediments dominated by terrestrial OM had lower methylation rates than those dominated by phytoplankton OM. The authors conclude from this result that plankton blooms stimulate mercury methylation and that sediments dominated by terrestrial OM have decreased bacterial activity that hampers mercury methylation (Figure 3). An alternative explanation is that the inorganic mercury was less bioavailable for methylation in sediments with high terrestrial OM because the presence of more organic matter (or OM quality) resulted in greater binding efficiency. Could the results be an artifact of the method, where the injected spikes of inorganic mercury were less bioavailable in the sediments dominated by terrestrial OM? This is an important point because a key conclusion of the study is that the lability of the OM for bacterial degradation is what controls the mercury methylation rate.

The reviewer raises an important point. Our results convincingly demonstrate higher bacterial activity in the sediments dominated by algal OM (Fig 2a and Supplementary Fig. 1). A greater binding efficiency of Hg to sediment with increased total carbon content has been observed for estuarine and marine sediments with comparably low total carbon content (typically from close to zero up to 12% loss on ignition)^{13,14}. In contrast the total carbon of our sediments was considerably higher, ranging from 12 to 32%. We lack the data to construct speciation models for Hg in the investigated sediment-porewater systems, but a modeling of conditions representative for boreal lakes (12-32% TOC, 35 mg/L DOC, 1 or 10 μM S(-II)(aq), 200 ng/g HgT, and assuming a thiol content in TOC and DOC of 0.15 mass% of carbon)¹⁵ predict that the partitioning of Hg is only affected to a minor extent by variations in total carbon content between 12 and 32%. The predicted log K_D is 4.4 and 4.6 at 12 and 32% total carbon, respectively for 1 μM S(-II)(aq), and 3.4 for both organic carbon contents at 10 μM S(-II)(aq). Regarding the potential effect of OM quality on Hg bioavailability, our results indicate that the variability in the total BP induced by differences in OM composition is the most important process for the observed variability in Hg methylation rate. However, our results also suggest that the OM is further affecting Hg methylation, either by additional specific activity enhancements of part of the microbial community which directly methylates Hg and/or by modulating Hg availability. The method used however does not allow us to disentangle between these two additional effects separately. Such discussion has been now included in lines 183–195.

6) Supplemental table 2 shows that the %methylmercury contents are

comparable in sediments dominated by both types of OM. Other studies have demonstrated that the %methylmercury content of sediments is a useful proxy for mercury methylation rate. Why is there a discrepancy in this dataset, where the sediments dominated by terrestrial OM have high methylmercury concentrations, high %methylmercury content but low methylation rates? The explanation provided is that the methylmercury in those sediments originates primarily from terrestrial/catchment sources. What evidence is available to support this assumption? Where measurements made of catchment loadings of methylmercury to those lakes? Hg loading can vary widely depending on catchment characteristics, hydrology and lake morphometry. More information is warranted to support their explanation.

Indeed the %MeHg/HgT has been used as a proxy for net MeHg formation under the assumption that the input-output of MeHg in the system are small enough to not alter the %MeHg/HgT ratio significantly^{16,17}. Some studies have demonstrated a correlation between k_m and %MeHg/HgT, which has been interpreted as that the %MeHg/HgT in the system is controlled by the Hg methylation rate, and not by MeHg demethylation or MeHg input-output. Such a correlation is also obtained in our data set of lakes dominated by autochthonous OM (Fig 2c). We would like to emphasize that we propose larger MeHg input from the catchment as an explanation for the statistically significant higher absolute concentration of MeHg in lake sediments dominated by terrestrial OM (p-value= 0.004). Previous studies have shown that terrigenous OM is the main vector for Hg and MeHg transport from catchment soils to surface waters in the boreal landscape³⁶. Further, the total Hg concentration was significantly higher (p-value= 0.006) in the sediment dominated by terrestrial OM (excluding the point source contaminated site Marnästjärn). This observation cannot be attributed to spatial differences in atmospheric Hg deposition¹⁸ but a consequence of higher terrestrial Hg input. These observations strongly support that there is also a higher input of MeHg from the catchment to the lakes where the sediment OM is dominated by terrestrial compounds. It is more intricate, and speculative, to attempt to explain the statistically not significant differences in %MeHg/HgT between the two groups of lakes (p-value = 0.79). As we unfortunately did not studied Hg neither %MeHg/HgT loads from the catchments, we did not make any attempt in the original manuscript to explain the differences in this parameter. However, considering that the imported OM from the catchment to the lakes is the main vector of MeHg for terrigenous lakes, high %MeHg/HgT percentages measured in these lakes indicates a substantial catchment formation and export of MeHg.

7) The discussion and analysis of dominant controls on mercury methylation in sediments is overly simplistic because there is little consideration of other environmental factors that have been previously demonstrated to play a key role such as pH, redox conditions, and temperature. The potential influences of those

other factors need to be acknowledged, in the very least in the discussion and even better in the statistical analysis.

As highlighted in the response to reviewers' comment 3, we agree with the reviewer on the relevance of environmental factors such as pH, redox, sulfide concentrations, temperature etc on Hg methylation processes in the environment. For example, Drott et al., 2007⁴ showed that while sulfide concentration might modulate the differences on k_m at different sediment depths, the "quality of OM" assessed by the CN ratio was the primary factor determining k_m . To address more specifically this question we compared two models: i) OPLS built with OM molecular composition of the surface sediment ii) OPLS model with the OM molecular composition of the surface sediment and the environmental factors measured in water overlying the sediment potentially known to affect Hg methylation processes (T, pH, O₂). The variability explained in k_m and power of prediction of the OPLS model built uniquely with molecular OM composition (R²Y= 86.6%; Q=71.2%) and the OPLS model built with OM plus O₂, T and pH were very similar (R²Y= 86.7%; Q=70.7%). Our results indicate therefore that OM molecular composition is the primary factor determining differences on Hg methylation rate constants across different boreal lake sediments. Changes in pH, redox, temperature and sulfur will likely modulate these rates between seasons¹⁹ or even sediment depths⁴. We have now acknowledged the effect of environmental parameters on lines 163–178 and in our concluding paragraph (lines 274–278).

8) Lines 214-216, page 9. The statement that methylmercury transported from terrestrial OM is more likely to be accumulate in the aquatic food chain is not broadly accepted in the field, and other studies have concluded that in situ methylmercury production is a greater source to the water column than catchment inputs (e.g., Watras et al. Environ Sci Technol. 2005 Jul 1;39(13):4747-58). The statement is misleading.

Indeed, the three reviewers asked about this, indicating that this point was not clearly addressed in our previous version. As it is not the focus on this work, we have decided to remove this discussion.

9) There is a contradiction in the emphasis and conclusions of the study. Lakes with sediment dominated by terrestrial OM showed lower mercury methylation rates but higher concentrations of methylmercury than sediment dominated with plankton OM, where higher mercury methylation rates were found. If this difference is explained by catchment loadings of methylmercury, then isn't the dominant factor controlling levels of methylmercury in sediments of the study lakes more related to the extent of catchment loading rather than the quality of the organic matter? Although mercury methylation rates were stimulated by planktonic blooms, the effect was less than that of external loading of

methylmercury. How relevant than is the quality of OM for controlling methylmercury accumulation in sediments.

We propose in the manuscript that for low productive lakes with high terrestrial OM inputs, the Hg and MeHg loads from catchment will be high and control the MeHg concentrations in sediments (as conceptually illustrated in Fig 4, right panel). Our results further show that, in contrast, in high productive lakes with less terrestrial OM loading, internal MeHg production (k_m) will be the primary source to MeHg concentrations in sediment. With the description of the combined OM molecular composition, we can therefore differentiate the processes and sources in control of MeHg concentrations in sediment between lakes governed by terrigenous OM inputs via runoff from those characterized by phytoplankton blooms. We believe we have now better explained the point raised by the reviewer in lines 247–255.

1. Graham, A. M., Aiken, G. R. & Gilmour, C. C. Dissolved organic matter enhances microbial mercury methylation under sulfidic conditions. *Environ. Sci. Technol.* **46**, 2715–23 (2012).
2. Graham, A. M., Aiken, G. R. & Gilmour, C. C. Effect of dissolved organic matter source and character on microbial Hg methylation in Hg-S-DOM solutions. *Environ. Sci. Technol.* **47**, 5746–54 (2013).
3. Bravo, A. G. *et al.* High methylmercury production under ferruginous conditions in sediments impacted by sewage treatment plant discharge. *Water Res.* **80**, 245–255 (2015).
4. Drott, A., Lambertsson, L., Björn, E. & Skjellberg, U. Importance of dissolved neutral mercury sulfides for methyl mercury production in contaminated sediments. *Environ. Sci. Technol.* **41**, 2270–2276 (2007).
5. Isidorova, A. *et al.* The effect of lake browning and respiration mode on the burial and fate of carbon and mercury in the sediment of two boreal lakes. *J. Geophys. Res. Biogeosciences* **121**, 233–245 (2015).
6. Schellekens, J., Buurman, P. & Pontevedra-Pombal, X. Selecting parameters for the environmental interpretation of peat molecular chemistry - A pyrolysis-GC/MS study. *Org. Geochem.* **40**, 678–691 (2009).
7. Nguyen, R. T. *et al.* Preservation of algaenan and proteinaceous material during the oxic decay of *Botryococcus braunii* as revealed by pyrolysis-gas chromatography/mass spectrometry and ¹³C NMR spectroscopy. *Org. Geochem.* **34**, 483–497 (2003).
8. Tolu, J., Gerber, L., Boily, J.-F. & Bindler, R. High-throughput characterization of sediment organic matter by pyrolysis–gas chromatography/mass spectrometry and multivariate curve resolution: A promising analytical tool in (paleo)limnology. *Anal. Chim. Acta* **880**, 93–102 (2015).
9. Kortelainen, P., Pajunen, H., Rantakari, M. & Saarnisto, M. A large carbon pool and small sink in boreal Holocene lake sediments. *Glob. Chang. Biol.*

- 10**, 1648–1653 (2004).
10. Verpoorter, C., Kutser, T., Seekell, D. A. & Tranvik, L. J. A global inventory of lakes based on high-resolution satellite imagery. *Geophys. Res. Lett.* **41**, 2014GL060641 (2014).
 11. Korthals, E. T. & Winfrey, M. R. Seasonal and spatial variations in mercury methylation and demethylation in an oligotrophic lake. *Appl. Environ. Microbiol.* **53**, 2397–404 (1987).
 12. Kellerman, A. M., Kothawala, D. N., Dittmar, T. & Tranvik, L. J. Persistence of dissolved organic matter in lakes related to its molecular characteristics. *Nat. Geosci.* **8**, 454–460 (2015).
 13. Hammerschmidt, C. R. & Fitzgerald, W. F. Sediment-water exchange of methylmercury determined from shipboard benthic flux chambers. *Mar. Chem.* **109**, 86–97 (2008).
 14. Driscoll, C. T. *et al.* Nutrient supply and mercury dynamics in marine ecosystems: A conceptual model. *Env. Res Lett* **119**, 118–131 (2013).
 15. Skjellberg, U. Competition among thiols and inorganic sulfides and polysulfides for Hg and MeHg in wetland soils and sediments under suboxic conditions: Illumination of controversies and implications for MeHg net production. *J. Geophys. Res.* **113**, G00C03 (2008).
 16. Drott, A., Lambertsson, L., Björn, E. & Skjellberg, U. Do potential methylation rates reflect accumulated methyl mercury in contaminated sediments? *Environ. Sci. Technol.* **42**, 153–158 (2008).
 17. Hammerschmidt, C. R. & Fitzgerald, W. F. Methylmercury cycling in sediments on the continental shelf of southern New England. *Geochim. Cosmochim. Acta* **70**, 918–930 (2006).
 18. Munthe, J., Wängberg, I., Rognerud, S. & Fjeld, E. *Mercury in Nordic ecosystems.* (2007).
 19. Hines, M. E. *et al.* Mercury methylation and demethylation in Hg-contaminated lagoon sediments (Marano and Grado Lagoon, Italy). *Estuar. Coast. Shelf Sci.* **113**, 85–95 (2012).

Reviewers' Comments:

Reviewer #1 (Remarks to the Author)

In re-reviewing this paper, I think the authors have done a reasonably good job addressing my review comments. The paper presents important results regarding the roles of terrigenous versus autochthonous OM in driving microbial processes in the study lakes. It is timely, addresses important environmental/ecosystem management issues, and is an important contribution to the field. I recommend publication after attending to a few minor editorial issues.

Minor comments

- 1.) Line 120-122. Without knowing sedimentation rates, the rates of microbial processes, or potential issues with sample collection, it is difficult to ascribe a reason for the similarities in the 2 sediment depths that were sampled. It is tough to get good resolution of sediment properties at the sediment water interface.
- 2.) Line 205-207 Actually, S₂₇₅₋₂₉₅, SR, and SUVA are all driven by DOM aromaticity. They indicate the presence of aromatic compounds in the sample (all based on absorption measurements in the UV). Phytoplankton derived DOM is low in aromatic compounds and therefore has a low SUVA value.

Overall, the paper needs some editorial assistance. I've noted a few items below, although I'm not a good editor:

Line 11 'of' rather than 'on'

Line 14 'in' rather than 'on'

Line 67 'compound' rather than 'compounds'

Line 87 'also presented' rather than 'presented also'

Line 89 'indicative of' or 'resulting from' rather than 'resulting of'

Line 117 'close' rather than 'closed'

Line 131 'separates' rather than 'separate'

Line 173 'while temperature' rather than 'while the temperature'

Line 190 'use' rather than 'uses'

Line 193 'by lower' rather than 'by a lower'

Reviewer #2 (Remarks to the Author)

The authors have made a thorough job at addressing the reviewers comments. I have no further concerns.

Reviewer #3 (Remarks to the Author)

I am satisfied overall with the clarifications and responses provided by the authors to my review comments. This manuscript contains novel research in the field of mercury biogeochemistry, with implications for understanding the environmental fate of mercury pollution in lakes. I recommend that this manuscript be accepted following a few minor revisions.

- 1) Temperature effects (comment #3 from reviewer #3): I am not fully convinced by the response of the authors that temperature was not a relevant influence on bacterial production and mercury methylation rates measured in the lake sediments. To address this point, I compiled the temperature data from Supplementary Table 1 and lake-mean bacterial production rates in sediments from Supplementary Table 2. There is a significant positive correlation between temperature and BP in the lakes (Pearson $r = 0.76$, $p = 0.012$, $n = 10$), and there is also a strong

negative correlation between temperature and lake depth (Pearson $r = -0.79$, $p = 0.006$, $n = 10$). Therefore, the eutrophic lakes, where autochthonous OM was dominant, were warmer and shallower systems. Although temperature may not have explained more variation in mercury methylation rate than OM composition in the OPLS model, the covariation between OM composition and lake temperature likely did not allow for a strong statistical test of temperature effects. I feel the authors should acknowledge that there were differences in depth and temperature regimes between the study lakes dominated by autochthonous OM and those dominated by terrestrial OM.

2) Line 13: Is it relevant to indicate in the abstract that the genetic bases (basis?) for bacterial methylation has been discovered?

3) Line 14: Change to "the role of organic matter composition"

4) Line 18-19: The boreal lakes dominated by terrestrial OM had far lower methylation rates but also lower demethylation rates, and so the net methylation rates (kmkd) were generally not lower. Isn't net methylation more relevant for determining the amount of methylmercury in sediments? I find this sentence in the abstract is a bit confusing and I suggest it be revised. For example, "In contrast, boreal lakes dominated by terrigenous organic matter inputs had far lower methylation rates but also lower demethylation rates. The higher concentrations of sediment methylmercury in systems dominated by terrigenous organic matter could not be accounted for by the net methylation rate suggesting an external source of methylmercury from the catchment was imported into those lakes via runoff."

5) Line 163 to 178: This discussion essentially dismisses the role of temperature in determining BP and methylation rate. As discussed in comment 1, the covariation of temperature regime and OM composition in the lakes should be acknowledged.

6) Line 179 to 195. The discussion in this paragraph could be shortened and tightened up, given the somewhat speculative nature of the explanations.

7) Line 196 to 227. This new paragraph is a large addition to the main body of the manuscript. I suggest that the conclusions be summarized in a few lines (e.g., line 222-227) and the bulk of the analysis be moved to the supplemental information.

8) Line 228-260. This paragraph is very important, in my opinion, and it could be edited to more clearly convey the main message that there is a decoupling of methylation rate and methylmercury concentrations in sediments dominated by terrigenous OM because of catchment inputs of methylmercury. It might be worth pointing out more directly that there is more methylmercury in terrigenous OM sediments for a given net methylation rate (supplement Fig 3a,b), which suggests an external source of methylmercury. It is also relevant to point out that although there were low mercury methylation rates in terrigenous OM sediments, there were also low methylmercury demethylation rates such that the net production of methylmercury was generally not lower.

9) Line 268: Change "higher MeHg levels in boreal lakes" to "higher MeHg levels in boreal lake sediments" because there are other processes (such as biodilution in eutrophic systems) that contribute to the fate of methylmercury in the food web.

Figure 3. Include temperature as a variable in the correlation analysis presented in this figure.

REVIEWERS' COMMENTS:

Reviewer #1 (Remarks to the Author):

In re-reviewing this paper, I think the authors have done a reasonably good job addressing my review comments. The paper presents important results regarding the roles of terrigenous versus autochthonous OM in driving microbial processes in the study lakes. It is timely, addresses important environmental/ecosystem management issues, and is an important contribution to the field. I recommend publication after attending to a few minor editorial issues.

Minor comments

1.) Line 120-122. Without knowing sedimentation rates, the rates of microbial processes, or potential issues with sample collection, it is difficult to ascribe a reason for the similarities in the 2 sediment depths that were sampled. It is tough to get good resolution of sediment properties at the sediment water interface. We agree with the reviewer. We have now removed the speculative part about the mixing processes. Please see new lines 131-134.

2.) Line 205-207 Actually, S275-295, SR, and SUVA are all driven by DOM aromaticity. They indicate the presence of aromatic compounds in the sample (all based on absorption measurements in the UV). Phytoplankton derived DOM is low in aromatic compounds and therefore has a low SUVA value. We agree with the reviewer that phytoplanktonic derived compounds have low SUVA. This is indeed in good agreement with our results. To avoid any misunderstanding, we have removed the lines 205-206 of the previous version. Please see new lines 234-246.

Overall, the paper needs some editorial assistance. I've noted a few items below, although I'm not a good editor:

Line 11 'of' rather than 'on'

Line 14 'in' rather than 'on'

Line 67 'compound' rather than 'compounds'

Line 87 'also presented' rather than 'presented also'

Line 89 'indicative of' or 'resulting from' rather than 'resulting of'

Line 117 'close' rather than 'closed'

Line 131 'separates' rather than 'separate'

Line 173 'while temperature' rather than 'while the temperature'

Line 190 'use' rather than 'uses'

Line 193 'by lower' rather than 'by a lower'

We thank the reviewer for the help on editing the manuscript. All the items detailed above have been now corrected according to the reviewer's suggestions.

Reviewer #2 (Remarks to the Author):

The authors have made a thorough job at addressing the reviewers comments. I have no further concerns.

Reviewer #3 (Remarks to the Author):

I am satisfied overall with the clarifications and responses provided by the authors to my review comments. This manuscript contains novel research in the field of mercury biogeochemistry, with implications for understanding the environmental fate of mercury pollution in lakes. I recommend that this manuscript be accepted following a few minor revisions.

1) Temperature effects (comment #3 from reviewer #3): I am not fully convinced by the response of the authors that temperature was not a relevant influence on bacterial production and mercury methylation rates measured in the lake sediments. To address this point, I compiled the temperature data from Supplementary Table 1 and lake-mean bacterial production rates in sediments from Supplementary Table 2. There is a significant positive correlation between temperature and BP in the lakes (Pearson $r = 0.76$, $p = 0.012$, $n = 10$), and there is also a strong negative correlation between temperature and lake depth (Pearson $r = -0.79$, $p = 0.006$, $n = 10$). Therefore, the eutrophic lakes, where autochthonous OM was dominant, were warmer and shallower systems. Although temperature may not have explained more variation in mercury methylation rate than OM composition in the OPLS model, the covariation between OM composition and lake temperature likely did not allow for a strong statistical test of

temperature effects. I feel the authors should acknowledge that there were differences in depth and temperature regimes between the study lakes dominated by autochthonous OM and those dominated by terrestrial OM. We agree with the reviewer that indeed the temperature might have enhanced the production of algal derived OM (planktonic blooms) and subsequently bacterial activity. Recent studies have demonstrated that the diversity of organic matter in boreal lakes is determined by climate (e.g. temperature) and hydrology (e.g. runoff inputs) (Kellerman *et al.*, 2014). High temperatures might be associated to higher *in situ* production of algal-derived OM and therefore to Hg methylation rate constants. We have now addressed the correlation of T with BP and with the presence on algal-derived OM in lines 189-194 of the revised version of the MS. We have also included the temperature in Fig 3.

2) Line 13: Is it relevant to indicate in the abstract that the genetic bases (basis?) for bacterial methylation has been discovered? We agree with the reviewer that the discovery of the genes involved in Hg methylation processes does not need to be in the abstract. We have corrected this sentence, please see lines 12-14.

3) Line 14: Change to “the role of organic matter composition”. We have modified the sentence according to this comment. See line 14.

4) Line 18-19: The boreal lakes dominated by terrestrial OM had far lower methylation rates but also lower demethylation rates, and so the net methylation rates (kmkd) were generally not lower. Isn't net methylation more relevant for determining the amount of methylmercury in sediments? I find this sentence in the abstract is a bit confusing and I suggest it be revised. For example, “In contrast, boreal lakes dominated by terrigenous organic matter inputs had far lower methylation rates but also lower demethylation rates. The higher concentrations of sediment methylmercury in systems dominated by terrigenous organic matter could not be accounted for by the net methylation rate suggesting an external source of methylmercury from the catchment was imported into those lakes via runoff.” We thank the reviewer for this suggestion. We find it difficult to introduce the concept of methylmercury demethylation within the word limit of the abstract and we believe that for the broad readership of *Nature Communications* there is a risk for confusion if demethylation is introduced already in the abstract without any explanation of the underlying processes. We therefore suggest that the discussion of demethylation is kept in the manuscript body text on lines 270-271, and we have modified the sentence in the abstract according to the reviewer's further comments (lines 19-20).

5) Line 163 to 178: This discussion essentially dismisses the role of temperature in determining BP and methylation rate. As discussed in comment 1, the covariation of temperature regime and OM composition in the lakes should be acknowledged. In agreement with the reviewer's remark, the discussion has been modified. Please see the response to the comment 1.

6) Line 179 to 195. The discussion in this paragraph could be shortened and tightened up, given the somewhat speculative nature of the explanations. We have shortened this part of the discussion from 236 to 186 words. See lines 247-262 of the revised version.

7) Line 196 to 227. This new paragraph is a large addition to the main body of the manuscript. I suggest that the conclusions be summarized in a few lines (e.g., line 222-227) and the bulk of the analysis be moved to the supplemental information.

As the reviewer mention this section was added during the revision process. The text paragraph was extended according to the previous comments 2, 4 and 6 of reviewer 1. We believe it is particularly important to highlight i) the good correlation of bulk parameters with the sedimentary OM molecular composition of the studied lakes and ii) the major improvement of incorporating a detailed molecular characterization of OM versus conventional parameters for the prediction of Hg methylation rates. Indeed, bulk parameters and optical properties of dissolved OM have recently been compared with the OM molecular

composition of dissolved OM in 109 boreal lakes (Kellerman *et al.*, 2015). Our study is however the first to show a correlation between bulk optical parameters in water overlying the sediment and the sediment OM composition. In order to include these two points in the MS and to integrate the comments of all reviewers in the final version of the manuscript, we have built a shorter version of this section in the MS by removing lines 202-206; 209-211; 216-222 of the previous version and my merging Lines 212-213 with line 211. Please see lines 246-263 of the new version.

8) Line 228-260. This paragraph is very important, in my opinion, and it could be edited to more clearly convey the main message that there is a decoupling of methylation rate and methylmercury concentrations in sediments dominated by terrigenous OM because of catchment inputs of methylmercury. It might be worth pointing out more directly that there is more methylmercury in terrigenous OM sediments for a given net methylation rate (Supplementary Fig 3a,b), which suggests an external source of methylmercury. It is also relevant to point out that although there were low mercury methylation rates in terrigenous OM sediments, there were also low methylmercury demethylation rates such that the net production of methylmercury was generally not lower. We have improved the flow of this section and highlighted the points raised by the reviewer. Please see lines 311-312; 357; 317-323 and 328-333 of the new version of the MS.

9) Line 268: Change “higher MeHg levels in boreal lakes” to “higher MeHg levels in boreal lake sediments” because there are other processes (such as biodilution in eutrophic systems) that contribute to the fate of methylmercury in the food web. We have modified the text according to reviewer comments. See line 3346.

Figure 3. Include temperature as a variable in the correlation analysis presented in this figure. The temperature has been now included in the Fig. 3.

References

Kellerman AM, Dittmar T, Kothawala DN, Tranvik LJ. (2014). Chemodiversity of dissolved organic matter in lakes driven by climate and hydrology. *Nat Commun* **5**: 3804.

Kellerman AM, Kothawala DN, Dittmar T, Tranvik LJ. (2015). Persistence of dissolved organic matter in lakes related to its molecular characteristics. *Nat Geosci* **8**: 454–460.